# Eliminating chemo-mechanical degradation of lithium solid-state battery cathodes during >4.5 V cycling using amorphous Nb₂O₅ coatings

Manoj K. Jangid [1,2,6], Tae H. Cho[1,6], Tao Ma [3], Daniel W. Liao [1], Hwangsun Kim[2,4], Younggyu Kim [1,2], Miaofang Chi [2,4] & Neil P. Dasgupta [1,2,5] ✉

Lithium solid-state batteries offer improved safety and energy density. However, the limited stability of solid electrolytes (SEs), as well as irreversible structural and chemical changes in the cathode active material, can result in inferior electrochemical performance, particularly during high-voltage cycling (>4.3 V vs Li/Li⁺). Therefore, new materials and strategies are needed to stabilize the cathode/SE interface and preserve the cathode material structure during high-voltage cycling. Here, we introduce a thin (~5 nm) conformal coating of amorphous $Nb_2O_5$ on single-crystal $LiNi_{0.5}Mn_{0.3}Co_{0.2}O_2$ cathode particles using rotary-bed atomic layer deposition (ALD). Full cells with $Li_4Ti_5O_{12}$ anodes and $Nb_2O_5$-coated cathodes demonstrate a higher initial Coulombic efficiency of 91.6% ± 0.5% compared to 82.2% ± 0.3% for the uncoated samples, along with improved rate capability (10x higher accessible capacity at 2C rate) and remarkable capacity retention during extended cycling (99.4% after 500 cycles at 4.7 V vs Li/Li⁺). These improvements are associated with reduced cell polarization and interfacial impedance for the coated samples. Post-cycling electron microscopy analysis reveals that the $Nb_2O_5$ coating remains intact and prevents the formation of spinel and rock-salt phases, which eliminates intra-particle cracking of the single-crystal cathode material. These findings demonstrate a potential pathway towards stable and high-performance solid-state batteries during high-voltage operation.

Motivated by the recent growth of electric vehicles (EVs), there is a significant worldwide effort focused on the development of safer and higher energy density solid-state batteries (SSBs)[1,2]. In SSBs, safety is enhanced by replacing the flammable liquid electrolyte (LE) with a non-flammable ceramic solid electrolyte (SE)[2,3]. Furthermore, SSBs

have the potential to enable high-energy-density electrode materials including Li-metal anodes and high-voltage cathodes[4–11].

Thiophosphate-based SEs, such as $Li_6PS_5Cl$ (LPSC), are one of the most commonly used SE materials because of their high ionic conductivity (>2 mS·cm⁻¹), formation of a kinetically stable SEI layer

¹Department of Mechanical Engineering, University of Michigan, Ann Arbor, MI, USA. ²MUSIC DOE Energy Frontier Research Center, University of Michigan, Ann Arbor, MI, USA. ³Michigan Center for Materials Characterization, University of Michigan, Ann Arbor, MI, USA. ⁴Center for Nanophase Materials Sciences (CNMS), Oak Ridge National Laboratory, Oak Ridge, TN, USA. ⁵Department of Materials Science & Engineering, University of Michigan, Ann Arbor, MI, USA. ⁶These authors contributed equally: Manoj K. Jangid, Tae H. Cho. ✉e-mail: ndasgupt@umich.edu

against Li metal, and malleable mechanical properties, which facilitate densification at room temperature[12–15]. However, in composite SSB cathodes that incorporate thiophosphate-based SEs, the cathode/SE interface tends to undergo oxidative chemical decomposition, particularly at high operating voltages (>4.3 V)[15–17]. Consequently, a thick and insulating cathode-electrolyte interface (CEI) is formed on the surface of cathode active material (CAM) particles, leading to a low initial Coulombic efficiency and increased cell polarization as a result of high interfacial impedance[14–17].

Among the available CAMs, the high-nickel-content layered $LiNi_xMn_yCo_zO_2$ (NMC) cathodes are associated with high theoretical Li-capacities (>275 mAh·g$^{-1}$), which can enable high energy densities, especially when cycled at elevated voltages (>4.3 V)[7,18–21]. However, during high-voltage cycling, several irreversible structural and chemical changes occur in NMC cathodes, including the formation of impurity phases (spinel and rock-salt), cation intermixing, oxygen evolution, and mechanical degradation such as intergranular or intraparticle cracking[6–10,19,20,22–25]. These changes, along with the unstable CAM/SE interface, contribute to a large increase in impedance and inferior electrochemical performance, including reduced rate capability and capacity fade during extended cycling. Recently, it has been shown that mechanical degradation (e.g., intergranular cracking) can be reduced by using single crystal (SC) NMC particles[4–10]. However, the interfacial instability challenges remain, and cracking still occurs when cycling at high-voltages (>4.3 V).

To improve the electrochemical performance of NMC CAMs, surface coatings and doping are two commonly employed methodologies. These approaches have been shown to reduce the undesirable interactions at the CAM/electrolyte interface, as well as helping to mitigate irreversible phase transformations and maintain the structural integrity of the CAM[14,21,26,27]. Surface coatings on NMC can lead to a more stable CAM/electrolyte interface and suppress unwanted interactions, resulting in improved Coulombic efficiency. Additionally, by doping a different element into the NMC crystal structure, cation intermixing can be suppressed and the transitional metal (T$_M$) layer spacing can be preserved[21,26].

The requirements for an ideal interfacial coating at the cathode/SE interface are multifold. First, the electronic conductivity must be sufficiently low, and the position of the band edges should be aligned to block undesirable charge transfer between the active material and the SE phase. Additionally, the area-specific resistance (ASR) associated with ionic transport through the coating must be sufficiently low to minimize cell polarization, which is amplified at higher current densities. This requires both a high ionic conductivity and small thickness. Additionally, the coating material must be chemically and electrochemically stable against both the SE and CAM phases over the range of voltages employed. Ideally, the coating would be chemically and structurally homogenous to avoid local current focusing at hot spots such as grain/phase boundaries, crystallographic defects, and spatial variations in chemical composition, topology, and grain orientation of the coating. The importance of chemical and structural homogeneity of the coating is often overlooked in the design of artificial SEI/CEI layers. However, recent studies have shown that amorphous coatings can enable fast-charging capabilities, which is attributed to a more uniform distribution of interfacial kinetics and transport compared to the composite natural SEI layer that forms based on electrolyte decomposition[28,29]. Hot spots can also arise if the coating is not perfectly conformal and continuous (pinhole-free), which requires precise synthesis methods such as atomic layer deposition (ALD)[13,28,30]. Finally, the coating must be sufficiently mechanically compliant to withstand the cyclic strains that occur in CAM particles during cycling.

Commonly used coating materials include metal oxides[30–34] and Li-containing metal oxides[17,27,34–39], Li borates[28], Li phosphates[40], etc. These coating materials, which act as an artificial CEI, have also been tested in composite SSB cathodes to explore the associated stability and performance improvements. Among these, Nb-based surface coatings, including $LiNbO_3$[17,27,34–37,41] have been widely explored for composite SSB cathodes. In addition to suppressing the CEI formation and providing mechanical support to the CAM particles, the $LiNbO_3$ interlayers/coatings act as a barrier to prevent oxygen release from the CAM particles during high-voltage cycling. However, to date, the reported stability of coated CAM materials at high voltages (≥4.5 V) in SSBs remains significantly lower than the performance required by EVs. Furthermore, the most common method of $LiNbO_3$ deposition on CAM particles employs solution coating followed by high-temperature annealing to form the polycrystalline $LiNbO_3$ phase, which can also result in Nb doping[5,33,41]. Therefore, there is need for new materials and fabrication methods to simultaneously achieve the requirements for an 'ideal' interlayer in SSB cathodes described above.

Owing to its fast Li kinetics, $Nb_2O_5$ could serve as an interesting coating material for CAM particles when used with sulfide-based SEs[42–44]. Interestingly, there is limited information available on the use of $Nb_2O_5$ as a coating material for high-voltage composite SSB cathodes. Aribia et al.[34] developed ALD NbO$_x$ coatings on sputter-deposited (planar) LiCoO$_2$ films and subsequently converted them into LiNbO$_x$ through a post-deposition thermal treatment, which demonstrated improved stability against a liquid electrolyte. ALD processes for direct coating of LiNbO$_x$ have also been reported[31], but these materials have not been shown to enable stable cycling at >4.5 V in a SSB cathode[35]. To date, there have been no reports of using amorphous $Nb_2O_5$ as an interlayer in composite SSB cathodes, which we demonstrate here for the first time.

In this study, we present the development of a thin (~5 nm) conformal amorphous $Nb_2O_5$ coating on single-crystal $LiNi_{0.5}Mn_{0.3}Co_{0.2}O_2$ (SC-NMC) cathode powder deposited by rotary-bed ALD. The composite SSB cathodes containing $Nb_2O_5$-coated SC-NMC particles show significantly improved electrochemical performance under high-voltage cycling (≥4.5 V vs Li/Li$^+$) including a higher initial Coulombic efficiency of 91.6% ± 0.5% compared to 82.2% ± 0.3% for the uncoated samples, improved rate capability (10x higher accessible capacity at 2C rate; 1C = 3 mA·cm$^{-2}$), and long-term cycling stability (99.4% after 500 cycles) compared to uncoated SSB cathodes. Using impedance analysis and post-cycling electron microscopy, these improvements in performance are shown to be attributed to a stable CAM/SE interface and the structural integrity of the layered CAM phase without intraparticle cracking. Overall, our findings provide valuable insights for the design of ALD-assisted controlled conformal coatings on bulk CAM phases for stable and high energy density SSBs, enabling outstanding cycling stability up to 4.7 V.

## Results and discussion
### ALD coating process and structural characterization
The procedure for depositing amorphous $Nb_2O_5$ coatings onto SC-NMC particles using ALD is depicted schematically in Figs. 1A and S1. ALD was performed on SC-NMC particles (sized 2–5 μm) without any additional pretreatment. To ensure conformal coverage of the entire particle surface without the presence of discontinuities at particle-particle contact points, a rotary-bed ALD reactor was used (Figs. 1A and S1)[45,46]. In this process, the cathode particles are constantly in motion and are suspended as they are agitated by the rotary bed system. In contrast, if artificial CEI coatings are formed on powders that are sitting on a substrate or in a crucible, the coating will form pinholes at the contact points, which will serve as hot spots for electrolyte decomposition.

The thickness, structure, and composition of the as-deposited $Nb_2O_5$ coatings were confirmed using transmission electron microscopy (TEM), X-ray Photoelectron Spectroscopy (XPS), and X-ray diffraction (XRD). TEM analysis and the associated fast Fourier transforms (FFTs) confirmed the presence of a ~5 nm thick amorphous coating in contact with the layered crystal structure (R-3m) of the

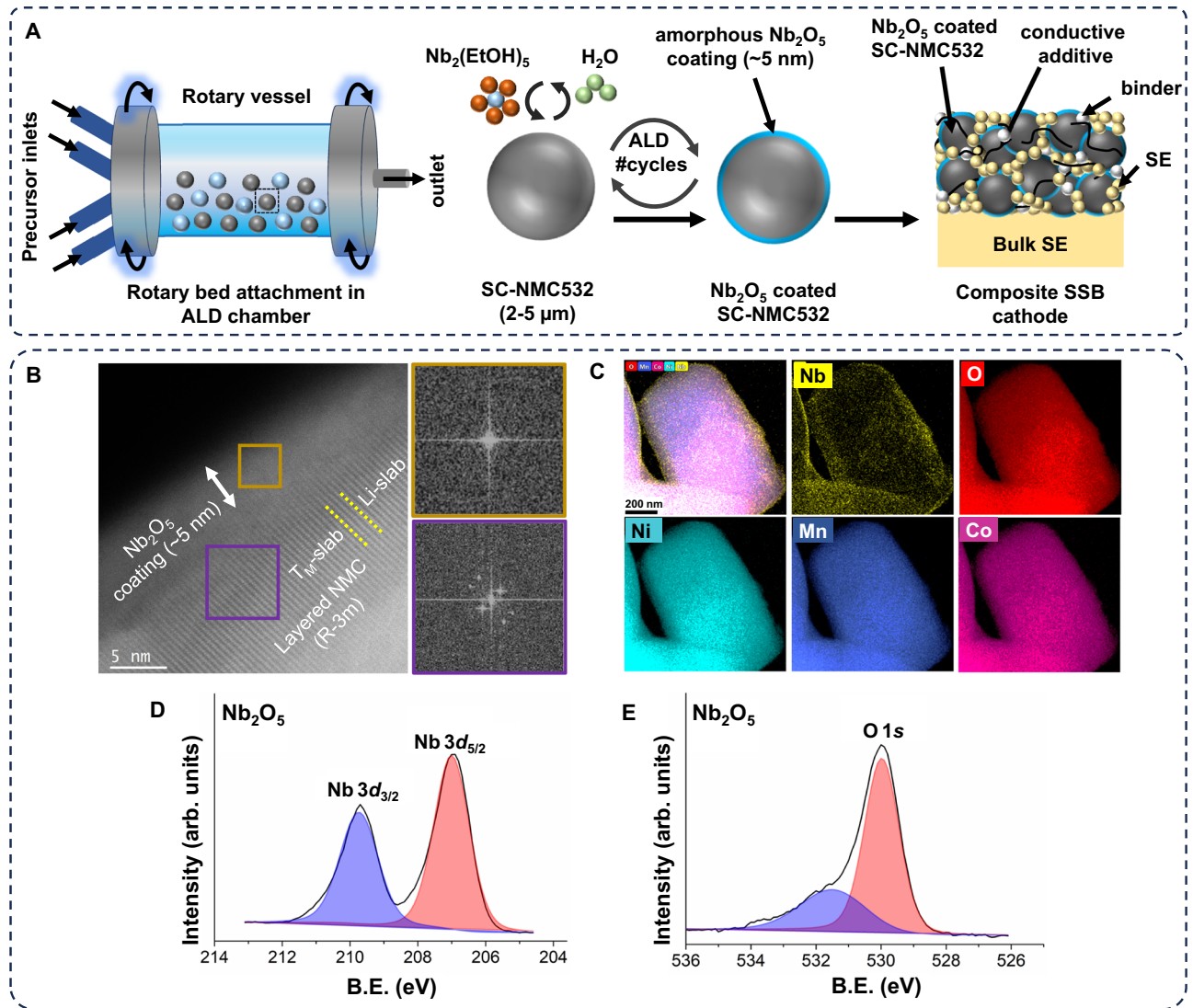

**Fig. 1 | Schematic and characterization of Nb₂O₅ coatings on SC-NMC cathode particles by rotary-bed ALD. A** Schematic of ALD equipped with a rotary-bed attachment, ALD process for depositing a 5 nm thick amorphous Nb₂O₅ coating on SC-NMC particles, and composite cathode assembly. **B** High-resolution TEM micrograph showing amorphous Nb₂O₅ coating and layered (R-3m) structure of an SC-NMC particle. FFTs from marked regions are also presented. **C** HDAAF-STEM EDS maps showing conformal Nb₂O₅ coating and distribution of elements in an SC-NMC particle. XPS core scans corresponding to (**D**) Nb 3d peak (**E**) O 1s peak from Nb₂O₅-coated SC-NMC powder.

SC-NMC (Fig. 1B). The elemental maps obtained through energy-dispersive X-ray spectroscopy (EDS) (Fig. 1C) further confirmed a uniform and conformal deposition of Nb₂O₅ coating on the NMC particles. The XPS and STEM-EDS results indicated an Nb:O ratio ≈2:5, corresponding to the Nb₂O₅ composition (Table S1 and Fig. S2). The Nb 3d core scan is consistent with the Nb⁵⁺ oxidation state (Fig. 1D)[43,44,47]. Furthermore, the O 1s core scan shows a strong peak at 530.0 eV, which is consistent with the Nb-O bond in Nb₂O₅, as well as a small shoulder peak (at 531.6 eV) associated with adventitious carbon, which is corroborated by the C 1s core scan (Figs. 1E and S3)[44]. Owing to the conformal and continuous nature of the ALD Nb₂O₅ coating on NMC particles, the Ni 2p, Co 2p, and Mn 2p core scans do not exhibit any peaks after coating (Fig. S4).

In this study, amorphous Nb₂O₅ films were deposited at a low temperature (175 °C), without any high-temperature post-annealing step. In contrast, most of the available literature on Nb-based coatings and doping in layered oxide cathodes[33–36,48,49] have reported that the development of Nb-based coatings typically entails a high-temperature (≥400 °C) annealing process, which may lead to Nb doping in addition to the formation of a crystalline Nb-based coating.

The extent and depth of Nb doping may further increase with increasing calcination temperature. Furthermore, high-temperature annealing often results in lithiation of NbOₓ surface films to form crystalline LiNbO₃.

The possibility of Nb doping of the NMC particles after ALD treatment in this study was therefore examined using XRD and XPS. Given that the ALD process for Nb₂O₅ occurs at a low temperature (175 °C), the probability of Nb doping is reduced significantly. XPS core scans for the Li 1s, Ni 3p, Mn 3p, Co 3p, and Nb 4s spectra (Fig. S5), as well as XRD scans of uncoated and Nb₂O₅-coated NMC samples, showed no noticeable shift in peak positions (Fig. S6)[27,48,50,51]. Similarly, XPS core scans of the Nb 3d and O 1s of the Nb₂O₅-coated NMC powder, with and without sputtering, revealed no discernible differences (Fig. S6), indicating the absence of Nb doping in the NMC structure. Collectively, these results highlight the power of rotary-bed ALD for conformal and pinhole-free modification of SC-NMC particles with ultra-thin amorphous Nb₂O₅ coatings, which would be difficult to achieve using alternative solution or line-of-sight deposition methods.

Composite cathodes consisting of uncoated and Nb₂O₅-coated SC-NMC particles were prepared by mixing Li₆PS₅Cl (as the SE),

graphitized carbon nanofibers (as the conducting additive), and PTFE (as the binder) in a weight ratio of 70:30:5:5 (or weight % ratio of 63.6:27.3:4.6:4.6). LPSC was selected as the SE phase in the composite cathode because of its favorable properties, including high ionic conductivity (>2 mS·cm⁻¹) and malleability (22 GPa)[11–13,52,53]. However, LPSC is known to be unstable at high voltages, which necessitates the use of artificial CEI layers. For the rate capability and long-term cycling stability tests, full cells were fabricated using composite anodes with $Li_4Ti_5O_{12}$ (LTO; 1.55 V vs Li/Li⁺) as the negative electrode material (Fig. S7 and details in "Methods" section)[54]. LTO was chosen as the anode material because of its flat voltage profile, zero strain during (de)lithiation, fast rate capability, and stable electrochemical potential against the LPSC SE[54]. In some of the electrochemical experiments, composite cathodes were also tested against a Li metal anode in a half-cell configuration. However, LTO anodes enabled us to focus on the performance of the NMC composite cathode without undesirable side effects from Li filament/dendrite propagation or void formation at the anode/SE interface[13,55–57].

The electrochemical performance of uncoated and $Nb_2O_5$-coated composite cathodes with an aerial capacity loading of 3 mAh·cm⁻² (based on the reversible capacities of 165 mAh·g⁻¹ at a voltage limit of 4.3 V vs. Li/Li⁺). The tests were conducted at three different upper voltage limits i.e., 4.3 V, 4.5 V, and 4.7 V vs Li/Li⁺, at 7 MPa stack pressure. Henceforth, the voltage values will be presented with respect to the Li/Li⁺ redox potential for clarity unless mentioned explicitly. We also note that when NMC cathodes are charged to higher voltages, the specific capacity increases. This results in a gravimetric capacity of 165 mAh·g⁻¹, 185 mAh·g⁻¹, and 205 mAh·g⁻¹ when cycled up to 4.3 V, 4.5 V, and 4.7 V vs Li/Li⁺, respectively. In other words, the areal capacity of the composite cathode when cycled to a 4.3 V, 4.5 V, and 4.7 V voltage limits will be 3.0 mAh·cm⁻², 3.36 mAh·cm⁻², and 3.73 mAh·cm⁻², respectively. Therefore, the LTO anode loading was adjusted based on the cut-off voltage to maintain a constant N/P ratio (further details in experimental methods and in Table S2).

### Effect on initial Coulombic efficiency (ICE)

Prior to conducting any specific electrochemical experiments, three formation cycles were performed on LTO|SE|NMC cells using a constant current and constant voltage (CCCV) protocol with a constant current (CC) of C/10 (where 1C = 3 mA·cm⁻²) and constant voltage (CV) hold until the current dropped to a value equal to C/25. The first cycle average CE of $Nb_2O_5$-coated cathode cells is significantly higher than that of similar uncoated cathode samples (Fig. 2). The $Nb_2O_5$-coated samples exhibited CEs of 91.6% ± 0.5%, 90.6% ± 0.3%, and 88.5% ± 0.2% at voltage limits of 4.3 V, 4.5 V, and 4.7 V vs Li/Li⁺, respectively. In contrast, the uncoated cathode samples showed CEs of 82.8% ± 0.6%, 82.2% ± 0.6%, and 79.8% ± 0.04% at the same voltage limits. Three samples of each type were tested at every voltage limit. The voltage traces during the first formation cycle of the uncoated samples exhibit higher cell polarization and appear more sloped, reaching the cut-off voltage earlier during the discharge cycle (Fig. S8).

In liquid electrolyte systems, the ICE of NMC cathodes is typically in the range of 83–86%[58–60] which is attributed to a combination of kinetic limitations and irreversible degradation processes during delithiation of the cathode[58–61]. This low ICE is one of the major challenges facing the current LIB technology, as ~15% of the battery capacity is lost during the initial cycles. Therefore, the observation of ICE values >90% for the coated SC-NMC particles in this study shows great potential for addressing one of the major challenges limiting current Li-ion battery technology. As we will show in the later sections of this study, the improved ICE is associated with a combination of improved electrochemical stability at the CAM/SE interface, as well as improved structural stability of the NMC particles. A detailed comparison of CEs during the formation cycles at different voltage limits is presented in Table S3.

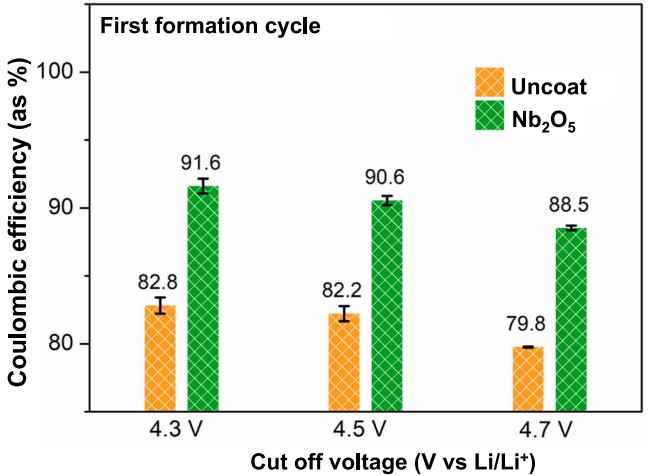

**Fig. 2 | Initial Coulombic efficiency of LTO|SE|NMC cells with uncoated and $Nb_2O_5$-coated cathodes cycled to different cut-off voltages during the first formation cycle.** The temperature during testing was 60 °C and stack pressure was 7 MPa.

### Effect on rate capability

The rate capability and corresponding voltage profiles of uncoated and $Nb_2O_5$-coated composite cathode samples after formation cycling are compared at different voltage limits of 4.3 V, 4.5 V, and 4.7 V (Figs. 3A–D and S9). The rate capability is tested at different current densities i.e., C/10, C/5, C/2, 1C, and 2C (where 1C = 3 mA·cm⁻²) using a constant current (CC) protocol. For the uncoated cathode (Fig. 3A), when cycled to 4.3V at a C/10 rate, the cell exhibits a discharge capacity of 2.63 mAh.cm⁻², which is reduced by 65% at 1C and 92% at 2C. After returning to a C/10 rate, the accessible capacity is fully recovered, indicating that this low-rate capability is associated with kinetic/transport limitations. When the uncoated samples are cycled at 4.5 V, although the initial accessible capacity at C/10 increases to 2.85 mAh·cm⁻² because of the higher cut-off voltage, the accessible capacity decreases by 72% and 95% at 1C and 2C rates, respectively. In the case of the uncoated samples cycled at 4.7 V, the accessible capacity at a C/10 rate is inferior to the previous two cases (2.45 mAh·cm⁻²), which is attributed to the lower ICE at 4.7 V. Furthermore, the cell cycled at 4.7 V exhibits a negligible capacity at faster rates (merely 2% and 0.2% of the C/10 capacity at 1C and 2C rates, respectively).

The poor rate performance observed in the uncoated samples during high-voltage cycling is consistent with prior reports[26,30,32,40,62]. The sulfide SE is unstable at high voltages (further details in Fig. S10)[14,15] and is oxidized by the CAM particles, leading to the formation of a thick CEI layer on the CAM particles[15–17]. In particular, when the cathode voltage is increased to above 4.3 V, oxygen release from the NMC cathode results in structural instability and increased degradation[6,7,16–20,63]. As a result, this thick CEI layer will result in reduced rate capability through a combination of high interfacial impedance and transport losses through the SE phase of the composite cathode.

On the other hand, the $Nb_2O_5$-coated cathode demonstrates superior rate capability (Fig. 3B). As discussed earlier, the $Nb_2O_5$ coating enables a high ICE during the formation cycles. Consequently, the initial accessible capacity of coated samples at a C/10 rate is higher than that of uncoated samples. The coated samples exhibit accessible capacities of 3.0 mAh·cm⁻², 3.1 mAh·cm⁻², and 3.52 mAh·cm⁻² during cycling at 4.3 V, 4.5 V, and 4.7 V, respectively. These values are all greater than 92% of the theoretical areal capacity at each voltage, illustrating the nearly full utilization of the coated CAM particles within the composite cathode.

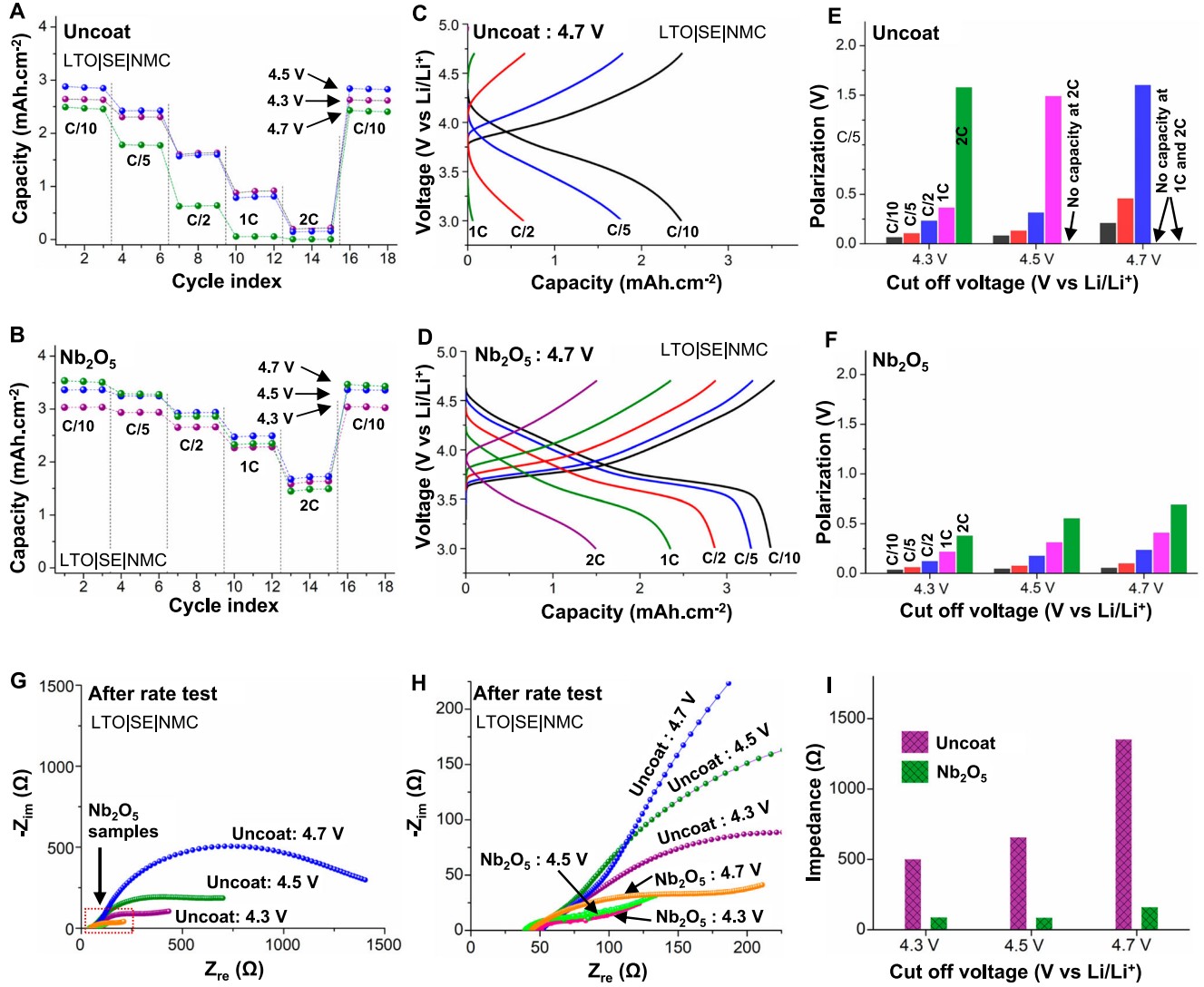

**Fig. 3 | Rate capability and impedance analysis for coated and uncoated cathodes.** Rate capability trends of LTO|SE|NMC cells with (**A**) uncoated and (**B**) $Nb_2O_5$-coated composite cathodes. Voltage profiles of (**C**) uncoated and (**D**) $Nb_2O_5$-coated cathodes at different C-rates (C/10, C/5, C/2, 1C, 2C; where 1C = 3 mA·cm⁻²) at a cut-off voltage of 4.7 V vs Li/Li⁺. Comparison of polarization estimated from dQ/dV analysis of voltage profiles at different C-rates for (**E**) uncoated and (**F**) $Nb_2O_5$-coated electrodes. **G** Nyquist impedance plots of uncoated and $Nb_2O_5$-coated cathodes after the rate capability tests. **H** Zoomed-in view showing Nyquist impedance plots of $Nb_2O_5$-coated cathodes. **I** Comparison of interfacial impedance for uncoated and $Nb_2O_5$-coated cathodes after the rate capability tests. The temperature during testing was 60 °C and stack pressure was 7 MPa.

Furthermore, the $Nb_2O_5$ coating has a more pronounced effect on rate capability at faster rates (1C and 2C). The coated samples show significantly higher capacity retention at a 2C rate. While the uncoated samples exhibit <8% retention of the initial C/10 capacity at 2C, the coated samples retained 54%, 51%, and 42% of the initial C/10 capacity when cycled at 4.3 V, 4.5 V, and 4.7 V, respectively.

To provide a comparison to state-of-the-art coatings, solution-processed LiNbO₃ coatings were also applied to SC-NMC cathodes[5,33,41]. The rate capability of ALD $Nb_2O_5$-coated cathodes cycled to 4.3 V and 4.7 V cut-off voltages is consistently higher than those of LiNbO₃-coated cathodes (Fig. S11). It is important to highlight that the solution processing method involves a high-temperature annealing step, resulting in a polycrystalline coating with microstructural heterogeneity. The superior performance of the $Nb_2O_5$ coating underscores the benefits of an amorphous coating.

We note that because the specific capacity of NMC532 increases as the cut-off voltage increases (i.e., 165 mAh·g⁻¹, 185 mAh·g⁻¹, and 205 mAh·g⁻¹ when cycled up to 4.3 V, 4.5 V, and 4.7 V, respectively), the applied current density for a specific C-rate during 4.5 V and 4.7 V

cycling is 1.12 times and 1.25 times the current density applied for the same C-rate at 4.3 V. The loading of the anode was adjusted accordingly to maintain a constant N/P ratio at each voltage. Therefore, the slightly lower capacity retention at 4.5 V and 4.7 V is likely attributed to a higher cell polarization (e.g., from a higher IR drop through the SE phase and increased anode thickness) at these higher current densities and is not indicative of increased interfacial degradation.

### Effect on cell polarization and impedance evolution

The effect of the $Nb_2O_5$ coating is also reflected in the voltage profiles of the samples (Fig. 3C, D and Figure S9). The voltage profiles of the uncoated samples exhibit a steeper slope relative to the coated samples, resulting in a lower accessible capacity. The polarization is estimated from the dQ/dV plots that were derived from the voltage profiles of the rate capability tests[64], as depicted in Fig. 3E, F and Figure S12. The dQ/dV analysis establishes that uncoated samples experience more polarization compared to the coated samples, and this effect worsens with increasing voltage limits and current densities.

To further study the influence of the $Nb_2O_5$ coating on the interfacial impedance, electrochemical impedance spectroscopy (EIS) measurements were performed at intermittent voltages during the first charge cycle and after the rate capability tests. The Nyquist impedance plots and fitted total impedance (excluding series resistance) of the uncoated and $Nb_2O_5$-coated electrodes at different voltage points during the first charge cycle, and after the rate capability tests at different voltage limits, are compared in Figure S13 and Fig. 3G–I. The fitted equivalent circuit is presented in Figure S14 and Table S4. As mentioned above, the interface between the uncoated CAM and the SE particles is unstable, resulting in the degradation of the SE and concomitant formation of a thick CEI layer on the CAM particle surfaces.

At lower voltages (≤4.3 V), the increase in impedance is primarily a result of this interphase formation. Beyond 4.3 V, irreversible and detrimental structural changes in CAM particles (as discussed later) further exacerbate the impedance. The combined effects of SE degradation, thick CEI formation, and irreversible structural changes within the CAM particles lead to a significant increase in the impedance that severely affects the Li transport across the CAM particles. Owing to the complex nature of the impedance contributions from the different interfaces (i.e., anode/SE and CAM/SE), here we report the total interfacial impedance.

To further probe the evolution of interfacial impedance as a function of voltage, intermittent EIS measurements were conducted during the first charging cycle. The interfacial impedance of the uncoated cathode consistently exhibits higher values (by 60%, 126%, 235%, and 164% at 3.9 V, 4.3 V, 4.5 V, and 4.7 V, respectively) with a more rapid increase beyond 4.3 V (Fig. S13). In contrast, the interfacial impedance of the $Nb_2O_5$-coated cathode is significantly lower and remains stable up to 4.5 V.

Similarly, after the rate test at 4.3 V, the interfacial impedance for the uncoated cathode is ~500 Ω. This value was significantly higher (~1350 Ω) for the uncoated cathode subjected to the rate test at 4.7 V. In contrast, for the $Nb_2O_5$-coated cathodes, impedance evaluation aligns well with their electrochemical performance and is notably lower compared to the similar uncoated cathodes. The interfacial impedance after the rate test at 4.3 V is 88 Ω (~5.7 times lower than that of the equivalent uncoated cathode). This increased to ~160 Ω for the cathode cycled at 4.7 V, which is approximately 8.5x lower than the similar uncoated cathode. The internal resistance for both uncoated and coated cathodes was also deduced from the voltage profiles obtained during galvanostatic intermittent titration technique (GITT) experiments, where a larger polarization and internal resistance were observed for the uncoated samples (Fig. S15)[64].

## Effect on long-term cycling stability

To assess the impact of the $Nb_2O_5$ coating on long-term cycling stability (after 500 cycles) of SSB composite cathodes, both the uncoated and coated electrodes (LTO|SE|NMC) were subjected to cycling at a 1C rate (3 mA·cm⁻²) at 4.5 V and 4.7 V voltage limits using a CC protocol (Fig. 4A). In the case of uncoated cathodes, the initial charge capacity was only 1.32 mAh·cm⁻² (equivalent to just 40% of the theoretical areal capacity), with a CE of 33% during the first cycle (after formation) at 1C and a 4.5 V limit, resulting in a discharge capacity 0.43 mAh·cm⁻². For the cells that were cycled to a 4.7 V limit, the accessible capacity and CE deteriorated further, resulting in a mere 0.60 mAh·cm⁻² initial charge capacity (16% of the theoretical areal capacity) and a CE of 5%, resulting in a discharge capacity of 0.03 mAh·cm⁻². Furthermore, during extended cycling the CE remains below 99% for the first 12 cycles with a 4.5 V limit, and for the first 300 cycles with a 4.7 V limit, indicating extended capacity fade (Fig. 4B). The poor performance of the uncoated electrode, particularly at high-voltage limits, can be ascribed to the detrimental and permanent interfacial and structural changes taking place at the CAM/SE interface and within the CAM particles that impede the reversible Li kinetics within the CAM particles, as described in detail in the next section.

The CE values observed in the early cycles shown in Fig. 4B for the uncoated sample are significantly lower than the ICE values during the formation cycles at 4.5–4.7 V (shown in Fig. 2). The reason for this low CE in the early cycles during 1C charging is as follows. After the formation cycles, the impedance of the uncoated cell increases significantly (Fig. S13), which is a result of the formation of a resistive CEI layer and rock-salt phase formation in the CAM particles. As a consequence, the accessible capacity at a 1C rate is significantly lower for the uncoated particles, as shown in the rate capability experiments (Fig. 3A, C, E). Analogously, during the extended cycling data shown in Fig. 4A, the accessible capacity during charging is significantly lower than during the first formation cycle, because a constant current 1C rate was used in Fig. 4A. Therefore, the additional losses in discharge capacity that occur during the early cycles of constant-current 1C charging at 4.5–4.7 V will represent a significantly larger fraction of the charge capacity than was observed during the C/10 CCCV formation cycle. In other words, because the denominator of the CE calculation (charge capacity) is significantly smaller at 1C compared to C/10, the CE value for a given loss in accessible capacity appears to be much larger.

In contrast, the $Nb_2O_5$-coated cathodes demonstrate superior charge capacities and CEs. These electrodes exhibited initial charge capacities of 2.63 mAh·cm⁻² and 2.81 mAh·cm⁻² at 4.5 V and 4.7 V limits, respectively. In other words, ~78% and ~75% of the theoretical electrode capacity can be accessed during CC charging at a high current density (1C) to a high voltage of 4.5 V and 4.7 V, respectively. This represents an increase by a factor of ~100% and ~365% compared to the initial charge capacities of the uncoated samples (Fig. 4A). Additionally, the CE of both the initial and extended cycles is higher for the $Nb_2O_5$-coated cathodes, resulting in reduced capacity fade (Fig. 4B). For example, the CE of the coated samples reaches above 99% by the second cycle for both the 4.5 V and 4.7 V limits and remains above 99.95% throughout the remaining 498 cycles. As a result, the discharge capacity has only decreased by 0.9% after 500 cycles at a 4.5 V limit and 0.6% after 500 cycles at a 4.7 V limit. A comparison of this performance to the other state-of-the-art Li-ion batteries cycled with both solid and liquid electrolytes to voltages ≥4.5 V is provided in Table S5, which illustrates that the remarkable stability observed in this study has not been previously achieved.

The long-term cycling stability of the $Nb_2O_5$-coated cathode was also assessed using a CCCV protocol at 1C with a 4.7 V limit (Fig. 4C). When a CCCV protocol is employed, the initial charge capacity increases to 3.41 mAh·cm⁻², which is 91.4% of the theoretical capacity. Additionally, the first cycle CE was 94% for a CCCV protocol, compared to 78% for the CC protocol (Fig. S16). However, during extended CCCV cycling, faster capacity fade occurs compared to CC cycling. Despite this fact, after 500 cycles, the accessible capacity was 2.47 mAh·cm⁻², which remains higher than the accessible capacity during CC cycling (2.18 mAh·cm⁻²). This represents a capacity retention of 72.4% of the initial accessible capacity after 500 cycles of CCCV charging to 4.7 V at 1C, which is unparalleled among previous reports on high-voltage cycling of NMC cathodes.

## Effect on structural stability of composite cathodes

The improved ICE, rate capability, and long-term cycling stability of the $Nb_2O_5$-coated electrodes, in comparison to the uncoated electrodes, can be attributed to the combined effects of a stable CAM/SE interface and the well-preserved structural integrity of CAM particles during electrochemical cycling at high-voltage limits. To corroborate this, a post-cycling analysis of cycled electrodes was conducted, encompassing investigations into the microstructure and phase evolution using high-angle annular dark-field scanning electron microscopy (HAADF-STEM), scanning electron microscopy (SEM), XPS, and XRD.

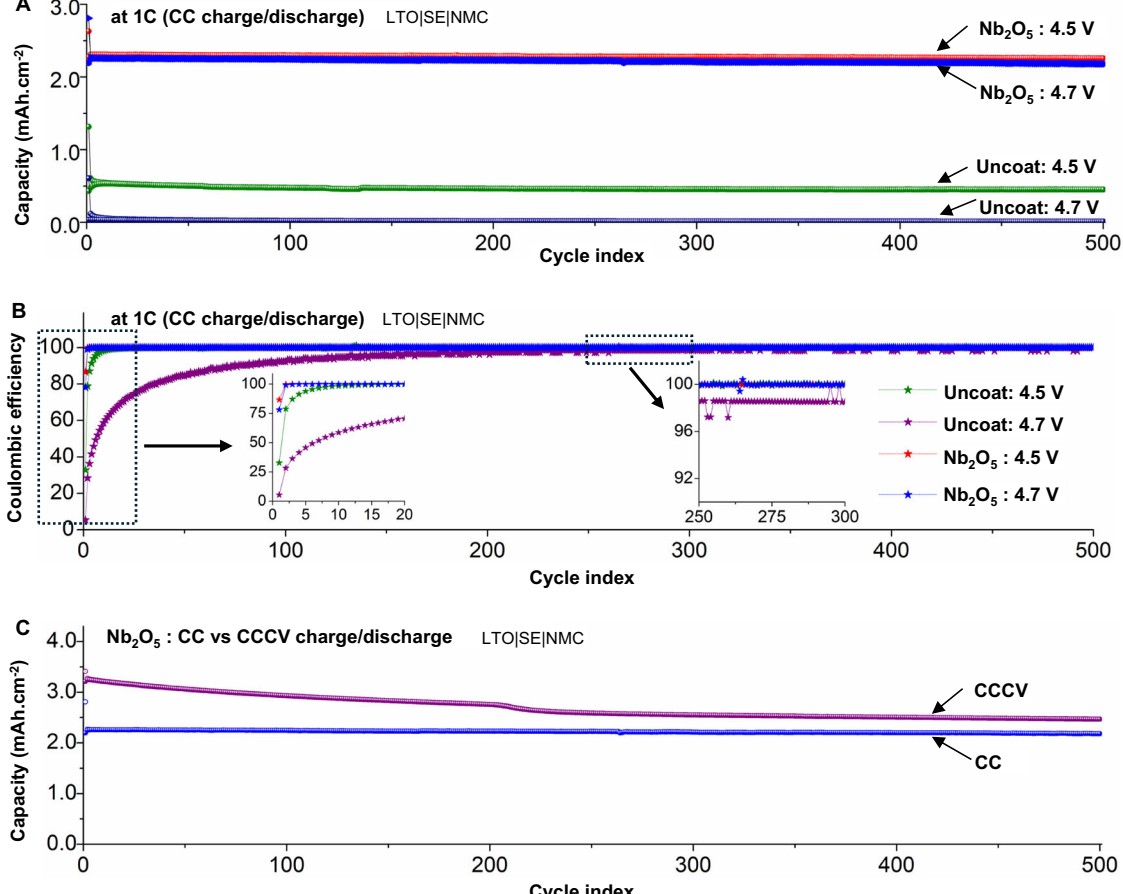

**Fig. 4 | Long-term cycling performance of cells with coated and uncoated cathodes.** **A** Long-term cycling stability and (**B**) associated Coulombic efficiencies of LTO|SE|NMC cells with uncoated and $Nb_2O_5$-coated SC-NMC cathodes at 1C rate (3 mA·cm$^{-2}$) cycled to 4.5 V and 4.7 V (vs Li/Li$^+$) voltage limits using a CC charge/discharge protocol. Insets in (**B**) show Coulombic efficiency during the first 20 cycles and for cycles 250 to 300. **C** Comparison of long-term cycling stability of $Nb_2O_5$-coated cathodes at 1C rate cycled to 4.7 V limit using CC (where a current equivalent to 1C is applied until the cell reaches the voltage limit) and CCCV (where a current equivalent to 1C is applied until the cell reaches the voltage limit followed by a voltage hold at the limit such that entire half cycle takes 1 h time) charge/ discharge protocols. The temperature during testing was 60 °C and stack pressure was 7 MPa.

The structural evolution of uncoated cathodes cycled to voltage limits of 4.3 V and 4.7 V after 500 cycles at 1C is elucidated through HAADF-STEM imaging of CAM particles accompanied with FFTs from the marked and zoomed regions (Fig. 5B, C) and with a schematic representation in Fig. 5A. Details of TEM sample preparation are outlined in the experimental method section. After 500 cycles using a 4.3 V limit, a rock-salt (Fm-3m) phase is observed in the CAM structure within a 2–5 nm proximity to the CAM/SE interface. This 'layered-to-rock-salt' transformation is attributed to the irreversible occupation of Li-sites, within the Li-slab, by the migration of transition metal (T$_M$) cations from the T$_M$-slab (green box region and FFT1 in Fig. 5B)[7,8,17–20,23–25,63]. Moving away from the rock-salt phase region towards the particle's interior, a distinct disordered phase is present (spinel, Fd-3m). This 'layered-to-spinel' transformation arises due to limited intermixing of T$_M$-cations with Li-sites in the Li-slab (purple box region and FFT2 in Fig. 5B)[7,8,17–20,23–25,63].

The migration of T$_M$-cations to the Li-sites within the Li-slab becomes significantly more pronounced when subjecting the uncoated cathode to high-voltage cycling (4.7 V), resulting in more adverse structural changes. The HAADF-STEM image of the uncoated electrode after 500 cycles using a 4.7 V limit is displayed in Fig. 5C. These images reveal the progression of the rock-salt phase deeper within the CAM particle, moving away from the particle's subsurface region. When layered cathodes are subjected to high-voltage charging (beyond 4.3 V), the rate of oxygen release is amplified, leading to oxygen

vacancies within the CAM particle. These vacancies, in turn, promote the formation of the rock-salt phase[7,17–20,23–25,63]. The FFTs obtained from regions, close to the particle surface (green box) and away from the surface (purple box), show a reduced interplanar distance of (~ 0.25 nm), in contrast to the layered phase (~0.48 nm), confirming the presence of the rock-salt phase.

These irreversible phase transformations are further confirmed by ex-situ XRD analyses conducted on composite cathodes after the rate capability tests at both 4.3 V and 4.7 V (Figs. 5D–F and S17). The XRD results obtained from uncoated cathodes cycled at 4.3 V reveal that the (003) peak and (018)/(110) doublet peaks exhibit an intermediate shift from their initial positions, while the (104) peak position remains unchanged. However, for the uncoated cathode cycled with a 4.7 V limit, the positions of the (003), (104) and (108)/(110) peaks exhibit a significant shift from their original positions. This phenomenon indicates a contraction of Ni–O and Ni–M interatomic distances due to the formation of the rock-salt phase, which possesses inferior Li-ion intercalation capabilities and sluggish Li kinetics[8,22,65]. Since the ex-situ XRD analysis was performed on samples after multiple high-rate cycles, the shift in XRD peak positions is expected to be permanent rather than residual[38,66,67].

In addition to the formation of detrimental phases (rock-salt and spinel), in HAADF-STEM images and FIB-SEM cross-section images, some of the uncoated CAM particles exhibit (sub)surface and intra-particle cracks (Fig. 6B–E), as illustrated in Fig. 6A. In the HAADF-STEM

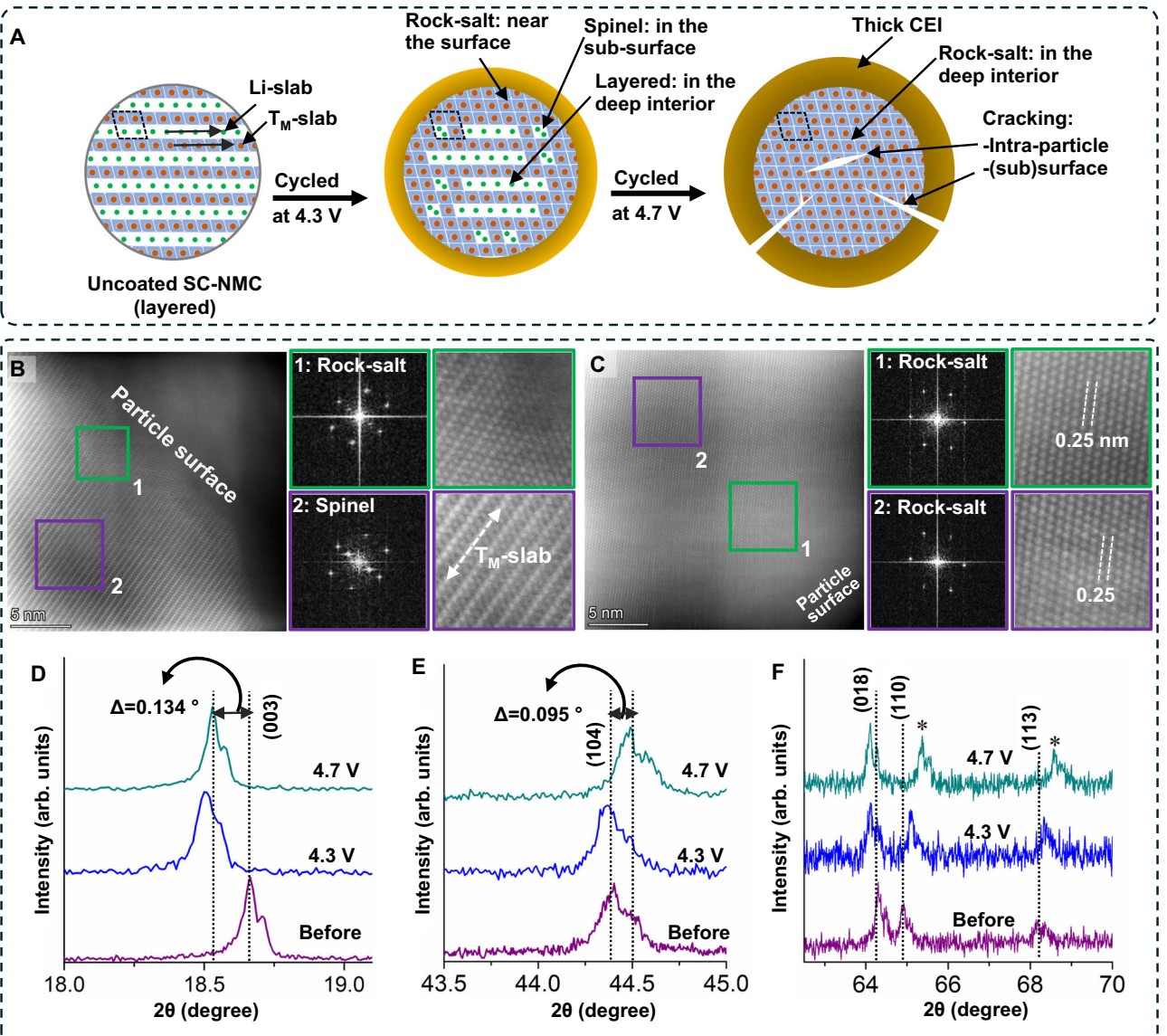

**Fig. 5 | Schematic and characterization of structural changes for uncoated SC-NMC particles after high-voltage cycling. A** Schematic showing structural changes occurring in uncoated SC-NMC532 particles when cycled to different voltage limits. Upon repeated cycling to high-voltage limits (>4.3 V vs Li/Li⁺), uncoated particles undergo irreversible structural changes (spinel and rock-salt phase formations), and (sub)surface and intra-particle cracking. HAADF-STEM images after 500 cycles at 1C rate (3 mA·cm⁻²) for (**B**) uncoated cycled at 4.3 V vs Li/Li⁺ and (**C**) uncoated cycled at 4.7 V vs Li/Li⁺ are presented. FFT patterns and zoomed-in views of marked regions are also presented as insets. (**D**–**F**) Ex-situ XRD scans of uncoated samples before cycling and from samples after rate capability tests at 4.3 V and 4.7 V limits.

images (Fig. 6B, C), the surface of the CAM particle displays a serrated morphology, featuring channel-like microcracks extending towards the particle's interior. With repeated cycling of the NMC cathode beyond 4.3 V, oxygen is released from the NMC crystal lattice resulting in heterogeneity (co-existence of different phases) and strain generation within the CAM particle, which is correlated with lattice gliding[68–72]. This chemo-mechanical behavior has been previously observed for single-crystal NMC cathodes in liquid electrolyte systems, where the lattice gliding and fracture behavior of individual particles is distinct compared to polycrystalline NMC particles, where inter-particle cracking commonly occurs[68–73]. As a consequence of these different fracture behaviors, single-crystal NMC cathodes have been shown to exhibit significantly lower amounts of oxygen gas release at high voltages than their polycrystalline counterparts[6,7,20]. In LE systems, particle cracking can be somewhat compensated by infiltration of the LE into the cracks, allowing for a continuous interface to be maintained. However, in SE systems, surface cracks will not be easily

infiltrated by the surrounding SE phase, resulting in a more severe drop in performance.

The oxygen release and concomitant lattice gliding become more prominent as the cut-off voltage increases, resulting in formation of microcracks in the (sub)surface regions along the Li-diffusion (003) direction[68–74]. The HAADF-STEM images shown in Fig. 6B and the associated FFTs demonstrate that the boundary regions of these microcracks exhibit the rock-salt phase (similar to the outer surface of the particle), whereas the interior regions that are further away from the crack boundaries are composed of the spinel phase[68–74]. This provides evidence for facile oxygen release from the newly exposed interior crack surfaces, where the required transport distance of oxygen is locally reduced. The co-existence of different phases results in compositional and structural inhomogeneity results in strain development within the CAM particles[22,62,68–76].

Upon repeated cycling at a high voltage (4.7 V) and high current density (1C rate), some of these microcracks gradually expand into

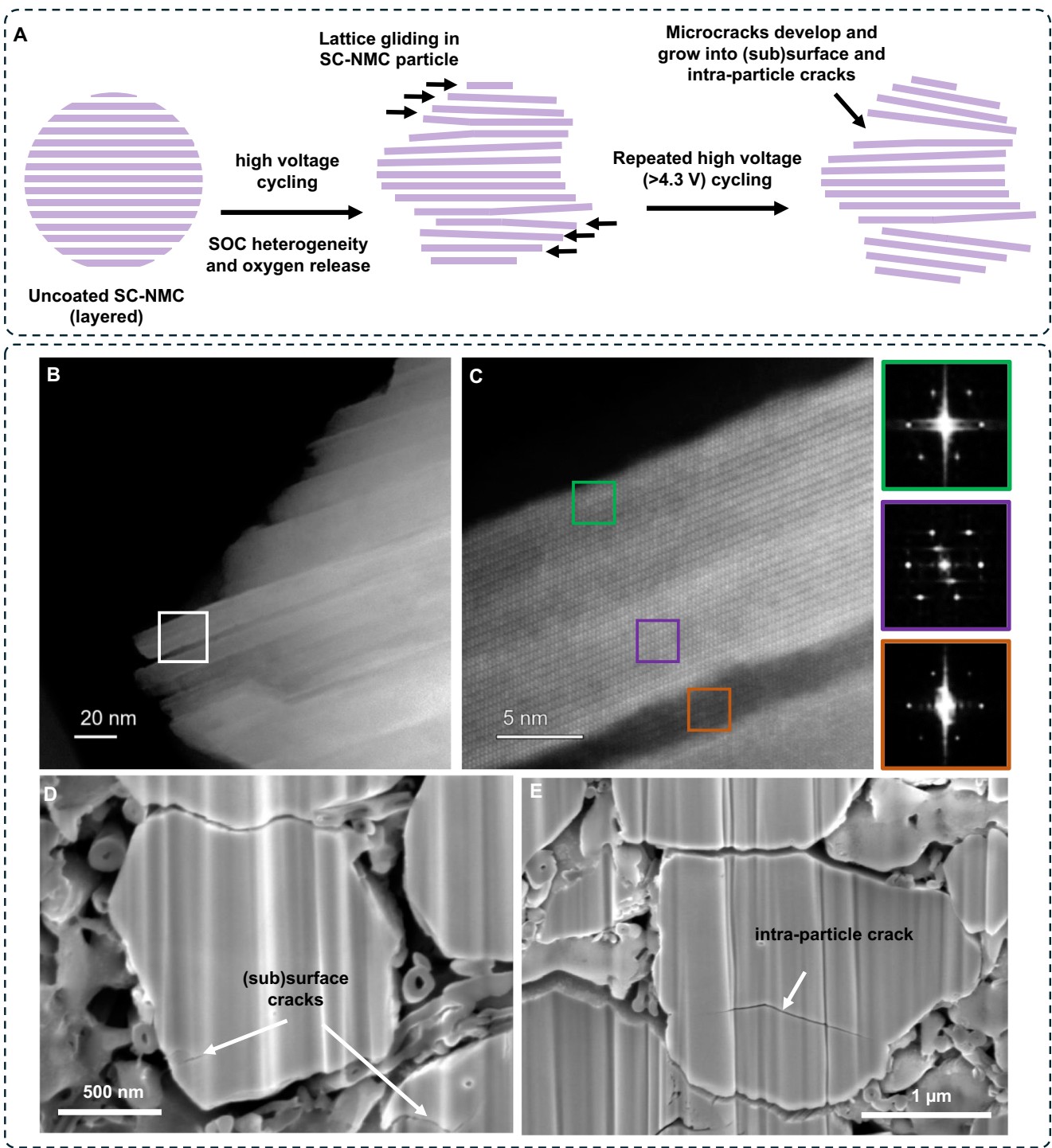

**Fig. 6 | Schematic and characterization of chemo-mechanical degradation in uncoated SC-NMC particles after high-voltage cycling. A** Schematic showing the onset of lattice gliding, subsequent microcrack formation, and its growth in (sub) surface and intra-particle cracking. **B** HAADF-STEM image of uncoated SC-NMC particle after 500 cycles at 1C rate (3 mA·cm$^{-2}$) and a voltage limit of 4.7 V vs Li/Li$^+$ shows a serrated surface resulting from lattice gliding and (sub)surface cracking. **C** Zoomed-in images of the cracks highlighted by the white box in (**B**) and FFTs from different marked regions. FIB-SEM cross-section images of uncoated SC-NMC electrode after 500 cycles at a 1C rate (3 mA·cm$^{-2}$) and a voltage limit of 4.7 V vs Li/Li$^+$ showing (**D**) (sub)surface and (**E**) intra-particle cracks.

larger fissures, ultimately manifesting as substantial intragranular cracks (Fig. 6E)[68–74]. This process might lead to the loss of contact between the CAM particles and SE phase, consequently impeding the supply of Li-ions. Overall, the collective impact of extensive electrolyte degradation, simultaneous thick CEI growth, coexistence of kinetically unfavorable phases, and CAM particle cracking, culminate in restriction in the Li kinetics/transport that results in a low ICE, limited

accessible capacity, and compromised rate capability for the uncoated composite SSB cathodes.

The structural evolution of Nb$_2$O$_5$-coated CAM particles after 500 cycles at 1C rate and 4.7 V limit is elucidated through HAADF-STEM imaging and ex-situ XRD (Figs. 7B–F and S17) and with a schematic in Fig. 7A. In contrast to the uncoated cathode, the layered (R-3m) structure of Nb$_2$O$_5$-coated CAM particles remains well preserved

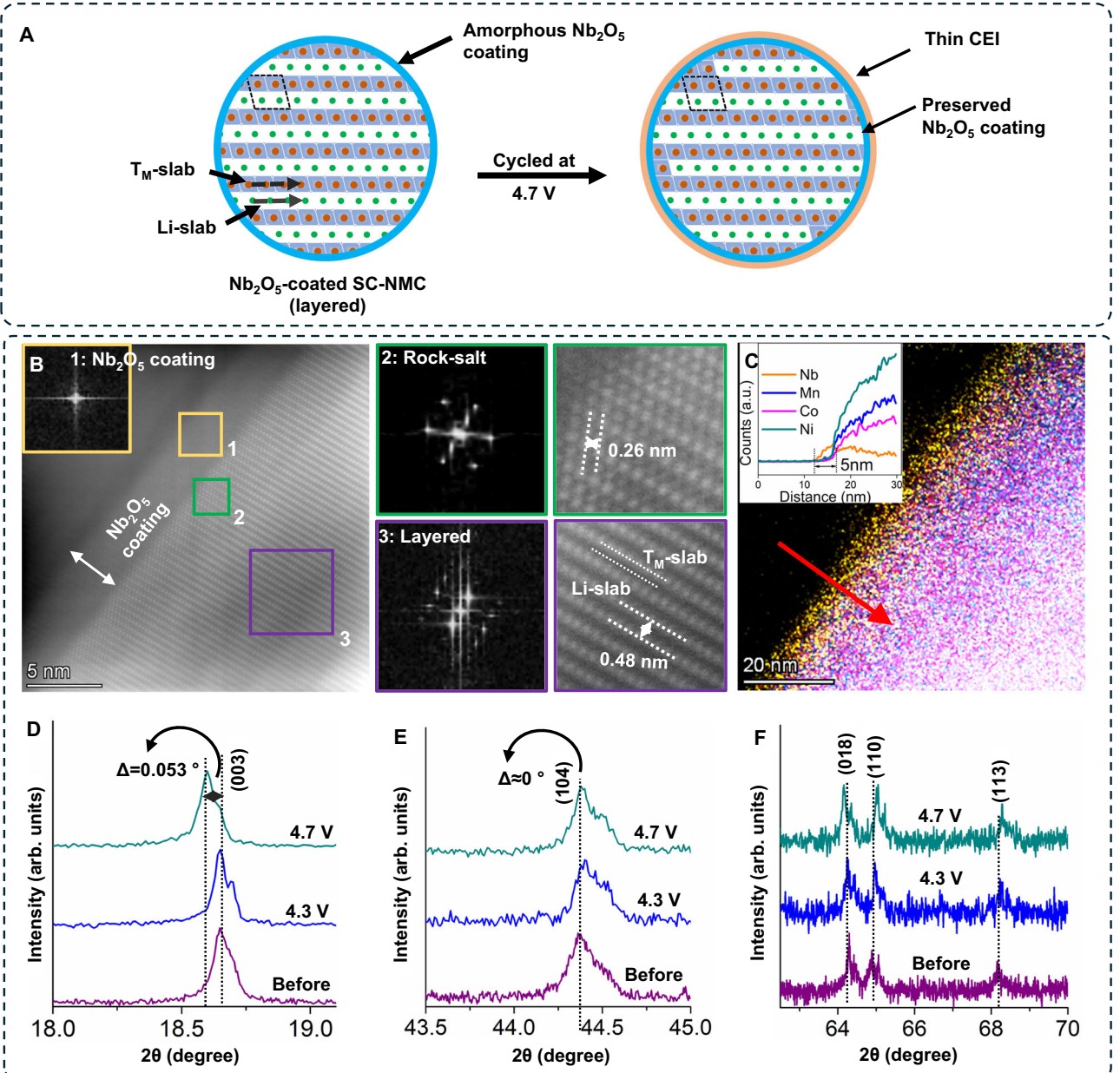

**Fig. 7 | Schematic and materials characterization of ALD-coated SC-NMC particles after high-voltage cycling. A** Schematic showing structural changes occurring in $Nb_2O_5$-coated SC-NMC532 particles when cycled at different upper voltage limits. For $Nb_2O_5$-coated NMC cathodes, both the NMC structure and $Nb_2O_5$-coating remain preserved even repeatedly cycled up to 4.7 V vs Li/Li$^+$. **B** HAADF-STEM images after 500 cycles at a 1C rate (3 mA·cm$^{-2}$) for $Nb_2O_5$-coated cathode cycled at 4.7 V limit. FFT patterns and zoom-in views of marked regions are also presented as insets. **C** STEM-EDS shows the preserved ~ 5 nm thick $Nb_2O_5$ coating and distribution of elements in an SC-NMC532 particle after 500 cycles at a 1C rate (3 mA·cm$^{-2}$) and a voltage limit of 4.7 V vs Li/Li$^+$. In the inset, the EDS line-scan shows the elemental distribution from particle surface to inside. (**D–F**) Ex-situ XRD scans of $Nb_2O_5$-coated samples before cycling and from samples after rate capability tests at 4.3 V and 4.7 V limits.

despite undergoing repeated cycling at high-voltage limits and elevated current density. Unlike the uncoated cathodes where a rock-salt phase forms throughout the CAM surface region, the presence of the $Nb_2O_5$ coating effectively inhibits the rock-salt phase formation. In fact, rock-salt phase development is limited and confined to a thin layer of merely 1-2 nm underneath the $Nb_2O_5$/CAM interface (FFT2 and green box region in Fig. 7B). Beneath this rock-salt region, a well-maintained layered structure (R-3m) with an interplanar spacing of 0.48 nm along the (003) direction is evident throughout the CAM particle volume (FFT3, purple box region in Figs. 7B, S18). The HAADF-STEM image (FFT1 and yellow box region in Fig. 7B) in combination with the EDS elemental line scan (Fig. 7C) and XPS analysis (Fig. S19)

affirms that even under the rigorous conditions of cycling at high-voltage limits and high current density, the $Nb_2O_5$ coating remains intimately adhered to the CAM particles, providing continuous protection to these particles.

The ex-situ XRD analysis (Figs. 7D–F and S17) of the cycled $Nb_2O_5$-coated cathodes further corroborates the TEM findings. In contrast to the uncoated cathodes, the positions of the majority of peaks in the XRD scans for the $Nb_2O_5$-coated cathodes before and after cycling either remain unchanged or exhibit only minimal shifts and no additional peaks are observed. These structural observations strongly support the enhanced electrochemical performance of the $Nb_2O_5$-coated cathodes presented in Figs. 2–4.

Owing to the improved phase stability of the $Nb_2O_5$-coated SC-NMC particles, the chemo-mechanical degradation that was observed for the uncoated particles is prevented when the coating is applied. To provide evidence for the improved mechanical stability, SEM analysis of the coated CAM particles after cycling to a 4.7 V was performed, and no particle cracking was observed throughout the electrode (Fig. S20). This illustrates the synergistic benefits of using both SC-NMC particles and the amorphous ALD coating to mitigate chemo-mechanical failure of CAM particles at high voltages. As a consequence of this improved stability, the SE/CAM particle interface will be more well-preserved, which is consistent with the improved rate capability and long-term cycling stability data of the coated particles (Figs. 2–4).

Furthermore, owing to the high coordination number of niobium and the high oxygen vacancy formation energy of $Nb_2O_5$,[77] the $Nb_2O_5$-coating is anticipated to effectively passivate the surface defects, such as surface oxygen vacancies, of CAM particles. If the oxygen vacancy formation energy of such TM-oxides significantly exceeds that of the CAM phase, the generation of oxygen vacancies within the TM-oxide layer would be impeded, leading to the formation of a barrier layer on the CAM particles. This barrier layer inhibits oxygen evolution and release from the CAM phase, thereby preserving the structural integrity and electrochemical stability of the layered cathodes at higher voltages[78–80]. The ability of the ALD coating to suppress both CEI formation and oxygen release at high voltages is uniquely enabled by the continuous and pin-hole-free nature of the amorphous $Nb_2O_5$ film. This illustrates the importance of using a rotary bed ALD system, to avoid pin-holes at the particle-particle contact points, which could otherwise act as localized sites for CEI formation and oxygen release.

In summary, the present study introduces the fabrication of a thin (~5 nm) amorphous $Nb_2O_5$ coating on single-crystal $LiNi_{0.5}Mn_{0.3}Co_{0.2}O_2$ layered cathode particles using an ALD reactor equipped with a rotary bed attachment. This attachment facilitates a uniform and pin-hole-free coating on CAM particles during ALD. The electrochemical performance of the resultant composite SSB cathodes is compared with uncoated electrodes cycled at different cut-off voltage limits. Through systematic post-cycling analyses involving impedance studies, structural evolution, and phase analysis, the study uncovers the underlying factors contributing to the differences in electrochemical performance between the uncoated and the $Nb_2O_5$-coated cathodes. The key findings are summarized as follows:

1. Composite SSB cathodes with $Nb_2O_5$-coated SC-NMC active materials particles exhibit an ICE of $91.6\% \pm 0.5\%$ at 4.3 V and $90.6\% \pm 0.3\%$ at 4.5 V vs $Li/Li^+$. These values are consistently ~9% higher than uncoated control samples, which results in an immediate 'savings' in capacity (and thus driving range for an EV application). Furthermore, this also addresses one of the major limitations of current state-of-the-art Li-ion battery technology, which suffers from low ICE (≤85%), where ~15% of the accessible capacity of the cathode is lost in the first formation cycle, even when they only charged to more modest voltages of ~4.2 V.

2. The composite SSB cathodes with $Nb_2O_5$-coated CAM particles exhibit improved accessible capacity, rate capability, long-term cycling stability, and reduced polarization compared to uncoated composite cathodes when cycled at voltages ≥4.5 V. For example, the $Nb_2O_5$-coated cathodes exhibited up to a 10x higher capacity at a 2C rate ($6\ mA·cm^{-2}$) compared to the uncoated cathodes. During long-term cycling at 4.7 V vs $Li/Li^+$, full cells (LTO|SE|NMC) exhibit a capacity retention of 99.4% over 500 cycles, with an average Coulombic efficiency >99.95%. The inclusion of the $Nb_2O_5$ coating significantly mitigates the SE degradation and preserves the chemical and structural stability of the CAM particles, which results in substantially lower impedance (a reduction of ~8.5x compared to uncoated cathodes after cycling at 4.7 V).

3. The uncoated cathodes undergo irreversible structural alterations. Upon repeated cycling at ≥4.3 V, the emergence of unfavorable rock-salt and spinel phases becomes evident near the CAM/SE interface and extends deeper into the particle interior. These phases hinder the facile Li-transport and re-intercalation within the CAM particles. Owing to the coexistence of different phases (rock-salt, spinel, and layered) and the occurrence of oxygen evolution beyond 4.3 V, the uncoated CAM particles experience chemo-mechanical stresses that result in lattice gliding. Upon repeated cycling at higher voltages, the lattice gliding prompts the formation of microcracks, which progressively culminate in the intra-particle cracking and irreversible structural deformation (rock-salt phase formation across the entire particle volume).

4. In the case of $Nb_2O_5$-coated CAM particles, even after the repeated cycling at 4.7 V, the occurrence of rock-salt and spinel phases remains limited and confined to a depth of 1–2 nm beneath the $Nb_2O_5$/CAM interface. Furthermore, throughout the majority of the particle volume, solely the electrochemically active layered phase is present. Notably, the indicators of lattice gliding, subsequent microcrack formation, and intra-particle cracking resulting from excessive oxygen release, as seen from the crystal structure of uncoated NMC particles, are absent in the $Nb_2O_5$-coated cathodes.

Overall, this study demonstrates a viable pathway to stable cycling of state-of-the-art single-crystal NMC cathode particles to voltages ≥4.5 V in a composite SSB cathode architecture. Moreover, the fundamental insights from electrochemical analysis and post-cycling electron microscopy reveal the critical chemo-mechanical degradation pathways of NMC cathodes in SSBs after high-voltage cycling, which are eliminated through the incorporation amorphous $Nb_2O_5$ coating. Finally, from a manufacturing perspective, the use of a rotary-bed powder ALD method enables complete and conformal coating of individual particles with an amorphous shell that is free of pinholes and microstructural heterogeneities, representing a potential pathway towards scalable manufacturing of high-performance SSB cathodes.

## Methods

### ALD process on SC-NMC532 powders

Atomic layer deposition (ALD) on SC-NMC532 powders (~2–5 μm particle size; MSE Supplies) was performed using a Savannah S200 ALD station with a rotary bed attachment. All depositions were performed at a substrate temperature of 175 °C. The $Nb_2O_5$ coating was achieved using Niobium (V) ethoxide (>99.9% purity, Strem Chemicals) and deionized water (DI water) as precursors. Ultra-high purity Ar (99.9999% purity) was used as the carrier gas at 70 sccm for the deposition process. The source temperature of Niobium (V) ethoxide was 110 °C. One ALD cycle consisted of Nb pulse (1s pulse), Ar purge (20s), DI water pulse (0.1s pulse), and Ar purge (20s). For all studies conducted in the manuscript, 30 ALD cycles were repeated to achieve a ~5 nm thick $Nb_2O_5$ coating on SC-NMC532 particles (as shown in Fig. 1B). The particle ALD reactor is schematically depicted in Fig. 1A. A detailed schematic of the ALD reactor with a rotary bed attachment and the ALD process is presented in Fig. S1. The rotary motor attached to the rotary bed was run at 3 rpm during the deposition process to conformally coat the NMC532 particle powder inside the tube. The base pressure of the system was 870 mTorr at 10 sccm.

### Solution-processed $LiNbO_3$ coating on SC-NMC532 powders

A 3 wt% $LiNbO_3$ coating was deposited on SC-NMC532 powders using a solution processing method[5,33,41]. A solution of 5% (w/v) lithium niobium ethoxide in ethanol (Thermo Scientific Chemicals; purity >99% metals basis) was added to 0.5 g of SC-NMC532 powder, stirred for 1 h, and dried at 70 °C for 12 h under Ar flow. Finally, this dried powder was annealed in an oxygen atmosphere at 425 °C for 1.5 h, resulting in a $LiNbO_3$ coating on SC-NMC532 particles.

## Composite electrode powder preparation

Composite cathode and anode powders were prepared inside an argon-filled glove box. For composite cathode preparation, NMC powders (uncoated or coated; Li-capacity 165 mAh·g$^{-1}$ at 4.3 V vs Li/Li$^+$), Li$_6$PS$_5$Cl SE particles (≤1 μm size; MSE Supplies; synthesized from >99.9% purity precursor materials), PTFE binder (Sigma Aldrich) and a graphitized carbon nanofiber conductive additive (Sigma Aldrich; purity >98% carbon basis) were mixed in a weight ratio of 70:30:5:5 (weight % ratio of 63.6:27.3:4.6:4.6). Composite anode powders were prepared by mixing Li$_4$Ti$_5$O$_{12}$ powder (Li-capacity 155 mAh·g$^{-1}$ vs Li/Li$^+$; MSE Supplies; purity >99%], Li$_6$PS$_5$Cl SE particles (≤1 μm size), PTFE binder, and a graphitized carbon nanofiber conductive additive in a 50:50:5:5 ratio.

## Cell fabrication

For LTO|SE|NMC full cell fabrication (Fig. S7), first, 0.0486 g of Li$_6$PS$_5$Cl SE powder (~10 μm size; MSE Supplies) was pressed into a 6 mm diameter die at a low load (100 MPa) for 10 seconds. Then, 0.0081 g of cathode composite powder (equivalent to 3 mAh·cm$^{-2}$ at 4.3 V vs Li/Li$^+$ cycling) was spread on one side of the SE green body and pressed at a 100 MPa load for 10 s. On the opposite side of the SE green body, LTO composite powder was spread, and finally, the entire LTO|SE|NMC stack was compacted at a 520 MPa load for 10 min. The n/p ratio was maintained at 1.1, by adjusting the anode loading according to the reversible capacity of the cathode at each cut-off voltage (further details in Supporting Information). The cathode composite powder loading was fixed for all samples (18.2 mg.cm$^{-2}$). The galvanostatic intermittent titration technique (GITT) experiments were conducted using Li|SE|NMC half-cells. For half-cells, a thin 6 mm diameter Li metal foil (~100 μm) was pressed on one side of the compacted SE|NMC pellets.

## Electrochemical testing

For electrochemical testing, the LTO|SE|NMC or Li|SE|NMC cells were assembled in a PEEK sleeve. Electrochemical testing was performed at 60 °C and 7 MPa stack pressure. The rate capability and long-term cycling stability tests were conducted using a Squidstat Plus potentiostat (Admiral Instruments). The lower voltage limit was set to 3.0 V vs Li/Li$^+$ (1.55 V as LTO vs Li/Li$^+$), and the upper voltage limits were set to 4.3 V, 4.5 V, and 4.7 V vs Li/Li$^+$. Prior to the rate capability and long-term cycling stability tests, three formation cycles were conducted using a CCCV (constant current and constant voltage) protocol where a constant current equivalent to C/10 (where 1C = 3 mA.cm$^{-2}$) rate was applied, followed by a CV hold at the respective voltage limits till the current dropped to a value equivalent to a C/25 rate. During the rate capability tests, three cycles were performed at each current density (i.e., C/10, C/5, C/2, 1C. 2C, and returning to C/10) using a CC protocol. The long-term cycling stability was performed at a 1C rate for 500 cycles. For the extended cycling tests using a CCCV protocol, a 1C rate was applied for the CC portion, followed by a CV hold until reaching a total charging time of 1 h. Electrochemical impedance spectroscopy (EIS) was performed in potentiostatic mode using a frequency range of 1 Hz and 1 MHz and V$_{rms}$ = 30 mV. The electrochemical instability of SE was investigated by cyclic voltammetry using a Li|Li$_6$PS$_5$Cl|SS (SS=stainless steel) blocking electrode within a voltage range of 0 to 4.7 V using a scan rate of 0.1 mV·s$^{-1}$ at 60 °C and 7 MPa stack pressure.

## Materials Characterization

Transmission electron microscopy was performed using a Thermo Fisher Talos F200X G2 S/TEM equipped with a Super-X EDAX detector. To prepare samples for TEM characterization, the composite cathode was carefully detached from the cycled LTO|SE|NMC cells and collected in Eppendorf vials. To remove the solid electrolyte phase, the detached cathode powders were washed multiple times with Ethanol and 2-propanol during sonication. Finally, the cleaned cycled SC-NMC powders were dispersed in 2-Propanol and drop-casted onto TEM grids, and left for drying overnight under an Argon atmosphere. The cross-sectional samples of the composite cathodes were prepared using plasma-focused ion beam-scanning electron microscopy (PFIB-SEM; Thermo Fisher Helios G4 PFIB UXe). X-ray diffraction was performed using a Rigaku SmartLab XRD instrument. XPS analysis was performed using a Kratos Axis Ultra with a monochromated Al Kα source that is directly integrated with an Argon glovebox to avoid any air exposure prior to analysis. The XPS data were analyzed with CasaXPS software. Cycled cathodes were carefully disassembled in an Ar glovebox and transferred to the XPS instrument without any air exposure.

## Data availability

All data generated and analyzed during this study are original and included in the article and the Supplementary Information. The data that support the plots within this paper and other findings of this study are available from the corresponding author upon request.

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

## Acknowledgements

All authors on this work acknowledge support from the Mechano-Chemical Understanding of Solid Ion Conductors (MUSIC), an Energy Frontier Research Center (EFRC) under Award No. DE-SC0023438 funded by the U.S. Department of Energy (DOE), Office of Science, Office of Basic Energy Science (BES). D.W.L. acknowledges support from the National Science Foundation Graduate Research Fellowship Program under Grant No. DGE-1256260.

## Author contributions

M.K.J. fabricated the electrodes and assembled cells, performed electrochemical cycling, conducted PFIB-SEM imaging, performed data analysis, and led the manuscript writing. T.H.C. performed the $Nb_2O_5$ ALD coatings. T.M. performed TEM characterization of post-cycling samples. D.W.L. performed the XPS analysis. H.K. and M.C. conducted TEM characterization of as-deposited and pristine NMC cathode powders. Y.K. assisted in the EIS analysis. N.P.D. supervised and conceptualized the research and assisted in manuscript writing and editing. All the authors participated in discussions during the manuscript writing and editing process.

## Competing interests

The authors declare no competing interests.
