## [Transparent Peer Review file · Nature Communications]

Eliminating chemo-mechanical degradation of lithium solid-state battery cathodes during >4.5V cycling using amorphous ALD coatings

Corresponding Author: Professor Neil Dasgupta

This manuscript has been previously reviewed at another journal. This document only contains reviewer comments, rebuttal and decision letters for versions considered at Nature Communications.

Version 0:

Reviewer comments:

Reviewer #1

(Remarks to the Author)

The authors have undertaken the suggested revisions, which has also involved additional work, and have also provided some arguments, which sound logical. Hence, in light of the significance of the results, as well as the extensive work done, I am happy to recommend publication of this manuscript in Nature Communications.

Reviewer #3

(Remarks to the Author)

This manuscript discusses the stability and performance enhancement of solid-state batteries (SSBs), specifically addressing the protection of the structural and chemical integrity of the cathode material of SSBs under high-voltage cycling (>4.5 V). As far as the reviewer is aware, the research topic has some value, but the content of the manuscript lacks some necessary analysis. Specific comments are given below. Based on these, I do not recommend this paper for publication. The authors also need to answer the following questions in the content of the manuscript, as commented below.

1. How did the atomic layer deposition (ALD) technique used in the study apply Nb₂O₅ coatings on single crystal NMC (LiNi_{0.5}Mn_{0.3}Co_{0.2}O₂) cathode particles? The authors should show more details on this.
2. The effect of monocrystalline and amorphous coatings on the charging rate needs to be elaborated by the authors.
3. What irreversible structural and chemical changes occur in the uncoated cathode materials mentioned in the article under high voltage cycling?
4. In the manuscript, the authors mention that it is the first time that Nb₂O₅ is used as a coating material for SSB cathodes with improved efficiency and cycling stability. On what technique were these results realized compared to other studies?
5. "Composite cathodes consisting of uncoated and Nb₂O₅-coated sc-NMC particles were prepared by mixing Li₆PS₅Cl (as the SE), graphitized carbon nanofibers (as the conducting additive), and PTFE (as the binder) in a ratio of 70:30:5:5." Is such a mixing ratio supported by the literature? Also, the proportions of the three materials are incorrectly indicated.
6. What are the results of the rate capability tests of the composite positive electrode at different voltage limits mentioned in the article, and how does the Nb₂O₅ coating affect this performance?
7. There are a large number of abbreviations in this paper that make reading and understanding of texts difficult. The parameters should be introduced in appendix section, their descriptions are omitted from the paper.
8. The analysis of the initial Coulomb efficiency was not specific enough. The results in Figure 2 are not sufficient to verify the high efficiency and stability of the Nb₂O₅ coating.
9. There are a large number of yellow-marked paragraphs in the main text content, which is not a professional representation.
10. The fonts of several images in the manuscript should all be appropriately enlarged to enhance readability.

Version 1:

Reviewer comments:

Reviewer #3

(Remarks to the Author)

The authors have revised the content of the manuscript in detail compared to the previous one, and the necessary analysis has enhanced the quality and rigor of the article. Therefore, this study on the stability and performance enhancement of solid-state batteries (SSBs) deserves to be published, provided that minor revisions are made to the following details. The specific comments are as follows.

1. The first occurrence of a term in the abstract should be explained in full, and the full name should be indicated in the SE to increase readability for the reader.
2. In the abstract, "Compared to uncoated samples at high voltages (≥ 4.5 V), the composite cathode with Nb₂O₅-coated CAM particles demonstrates a high initial Coulombic efficiency (91% vs. 82%)," the comparative representation of Coulombic efficiency in parentheses seems inappropriate, and a more standardized expression is suggested.
3. Units appearing in the manuscript should be examined in detail. For example, should 2 mS.cm⁻¹ be revised to 2 mS·cm⁻¹, and should >275 mAh.g⁻¹ be revised to >275 mAh·g⁻¹?
4. "Recently, it has been shown that mechanical degradation (e.g., intergranular cracking) can be reduced by using single crystal (sc) NMC particles". Why is the abbreviation single crystal used here with a lowercase sc?
5. "This last requirement is often overlooked in the design of artificial SEI/CEI layers; however, it has recently been shown that amorphous coatings have the potential to enable fast charging rates". This sentence suggests a change.
6. "The composite SSB cathodes containing Nb₂O₅-coated sc-NMC particles, along with LPSC solid electrolyte, binder, and conducting additive, show significantly improved electrochemical performance under high voltage cycling (≥ 4.5 V) including initial Coulombic efficiency (CE. 91% vs. 83%)" where sc and CE; 91% vs. 83% are suggested to be modified.
7. The font size in Figure 4 is not consistent. For example, LTO/SE/NMC.

August 8, 2024

RE: Response to reviewer comments for manuscript entitled: Eliminating chemo-mechanical degradation of solid-state battery cathodes during >4.5V cycling using amorphous ALD coatings by Manoj K. Jangid, Tae H. Cho, Tao Ma, Daniel W. Liao, Hwangsun Kim, Younggyu Kim, Miaofang Chi, Neil P. Dasgupta.

Dear Editorial Office,

We thank the reviewers and greatly appreciate their valuable feedback. We have carefully addressed their comments through additional experiments, new analysis, and relevant references. Changes to the manuscript have been highlighted in the revised manuscript, SI, and response letter document below. Thank you for your consideration of this manuscript for publication in *Nature Communications*.

Reviewer #1:

The authors have undertaken the suggested revisions, which has also involved additional work, and have also provided some arguments, which sound logical. Hence, in light of the significance of the results, as well as the extensive work done, I am happy to recommend publication of this manuscript in *Nature Communications*.

Response: We thank the reviewer for their supportive comments and for recommending publication of our research in *Nature Communications*. Your thorough feedback was invaluable in enhancing the quality of our manuscript.

Reviewer #3:

This manuscript discusses the stability and performance enhancement of solid-state batteries (SSBs), specifically addressing the protection of the structural and chemical integrity of the cathode material of SSBs under high-voltage cycling (>4.5 V). As far as the reviewer is aware, the research topic has some value, but the content of the manuscript lacks some necessary analysis. Specific comments are given below. Based on these, I do not recommend this paper for publication. The authors also need to answer the following questions in the content of the manuscript, as commented below.

Response: We thank the reviewer for their valuable feedback on our manuscript, and for commenting that the research topic has value. We have carefully considered the reviewer's specific comments and in response, we have provided detailed answers and have added significant new experiments, analysis, and discussion to the manuscript. We feel that this new content significantly increases the impact and depth of analysis in the paper.

Comment 1: How did the atomic layer deposition (ALD) technique used in the study apply Nb₂O₅ coatings on single crystal NMC (LiNi_{0.5}Mn_{0.3}Co_{0.2}O₂) cathode particles? The authors should show more details on this.

Response: We thank the reviewer for the comment. Unlike conventional ALD processes, which are performed on “static” substrates that are not moving, one of the important and novel aspects of this work is that the cathode powders were coated in a ‘rotary bed’ ALD mode. In this process, the cathode particles are constantly in motion and are suspended as they are agitated by the rotary-bed system. This is analogous to a clothes dryer, where the rotational motion of the machine causes the clothes to “tumble”, exposing surface area and accelerating the drying process.

The rotary-bed ALD process was originally developed for powder metallurgy and ceramic engineering [*Surf. Coat. Technol.* **213** (2012) 183-191; *J. Vac. Sci. Technol. A* **38** (2020) 052403; *J. Vac. Sci. Technol. A* **25** (2007) 67–74], and its use for solid-state battery cathode particles is a novel aspect of this work. The cathode powder is mixed with ZrO₂ balls and is placed in a tube inside the rotary vessel that is attached to a rotary motor. The rotary motor rotates the vessel, ensuring a conformal ALD coating on every particle in the powder.

In the ALD coating of powders, this agitation is critical to ensure that the ALD films are completely conformal along the entire particle surface. In contrast, if the powder bed was stagnant, there would be discontinuities/pinholes in the coatings at the particle-particle point contacts. For battery cathodes, these discontinuities become sites for unwanted surface reactions upon contact with electrolyte (SEI formation) and gas release at high voltage.

A detailed description of the ALD process parameters is provided in the 'Methods' section. To improve the clarity of the rotary-bed ALD process, in addition to Figure 1A, we have added a new detailed schematic of the ALD reactor equipped with a rotary bed attachment and the ALD process (Figure R1 below) in the revised SI as Figure S1. Briefly, one ALD cycle consists of an Nb pulse (1s), an Ar purge (20s), a DI water pulse (0.1s), and an Ar purge (20s). Thirty ALD cycles were repeated to achieve a ~5 nm thick Nb₂O₅ coating on sc-NMC532 particles. The deposition was performed at a substrate temperature of 175 °C, avoiding the formation of any crystalline LiNbO_x coating or Nb doping, as discussed in the manuscript.

The following new Figure has been added to the SI as Figure S1:

Figure S1: (A) Schematic of an ALD chamber with a rotary-bed attachment for conformal ALD coating at the particle scale on powders. (B) Schematic of the ALD process of Nb_2O_5 coating on an individual single-crystal NMC532 particle in the ALD chamber.

To provide further context for the powder ALD process, and its advantages compared to traditional (static) ALD modes, the following discussion and references have been added to the main text:

The procedure for depositing amorphous Nb_2O_5 coatings onto sc-NMC particles using ALD is depicted schematically in Figure 1A and Figure S1. ALD was performed on sc-NMC particles (sized 2-5 μm) without any additional pretreatment. To ensure conformal coverage of the entire particle surface without the presence of discontinuities at particle-particle contact points, a rotary bed ALD reactor was used (Figure 1A and Figure S1)^{45,46}. In this process, the cathode particles are constantly in motion and are suspended as they are agitated by the rotary-bed system. In contrast, if artificial CEI coatings are formed on powders that are sitting on a substrate or in a crucible, the coating will form pinholes at the contact points, which will serve as “hot spots” for electrolyte decomposition.

45. McCormick, J. A., Cloutier, B. L., Weimer, A. W. & George, S. M. Rotary Reactor for Atomic Layer Deposition on Large Quantities of Nanoparticles. *J Vac Sci Technol A* **25**, 67–74 (2007).

46. Longrie, D., Deduytsche, D., Haemers, J., Driesen, K. & Detavernier, C. A Rotary Reactor for Thermal and Plasma-Enhanced Atomic Layer Deposition on Powders and Small Objects. *Surf Coat Technol* **213**, 183–191 (2012).

In addition, the following text has been added to the methods section, to point to this new schematic:

A detailed schematic of the ALD reactor with a rotary bed attachment and the ALD process is presented in Figure S1

Comment 2: The effect of monocrystalline and amorphous coatings on the charging rate needs to be elaborated by the authors.

Response: We thank the reviewer for this comment and agree that it is important to clearly explain the benefits of using single-crystal particles and amorphous coatings in this study. In fact, it is the synergistic combination of these two characteristics (single-crystal vs. polycrystalline cathode particles, and amorphous vs. polycrystalline coatings) that distinguishes our work from previous efforts and enable the improved chemo-mechanical stability observed.

First, we will describe the importance of using single-crystal (sc) NMC in this study. As first described in the introduction section:

Recently, it has been shown that mechanical degradation (e.g., intergranular cracking) can be reduced by using single crystal (sc) NMC particles^{4–10}.

This ability to eliminate intergranular fracture is one of the major motivations to use sc-NMC particles in this study. To further clarify this point, the following text was added to the discussion:

This chemo-mechanical behavior has been previously observed for single crystal NMC cathodes in liquid electrolyte systems, where the lattice gliding and fracture behavior of individual particles is distinct compared to polycrystalline NMC particles, where interparticle cracking commonly occurs^{68–73}. As a consequence of these different fracture behaviors, single crystal NMC cathodes have been shown to exhibit significantly lower amounts of oxygen gas release at high voltages than their polycrystalline counterparts^{6,7,20}.

We have further emphasized this point in showing how both sc-NMC and the amorphous coating provide a synergistic benefit to avoid chemo-mechanical degradation:

To provide evidence for the improved mechanical stability, SEM analysis of the coated CAM particles after cycling to a 4.7 V was performed, and no particle cracking was observed throughout the electrode (Figure S20). This illustrates the synergistic benefits of using both sc-NMC particles and the amorphous ALD coating to mitigate chemo-mechanical failure of CAM particles at high voltages. As a consequence of this improved stability, the SE/CAM particle interface will be more well preserved, which is consistent with the improved rate capability and long-term cycling stability data of the coated particles (Figures 2,3,4).

The use of amorphous, as opposed to polycrystalline, coatings in this study is another novel and critical factor that differentiates our work from previous studies that have primarily focused on the crystalline LiNbO_3 phase. The amorphous coatings are chemically and structurally homogeneous. As a result, at the nanoscale, the coatings are free of current-focusing 'hot spots', which results in a more uniform distribution of interfacial kinetics and transport compared to polycrystalline coatings. The amorphous coatings are also mechanically compliant and are able to withstand the cyclic strains that occur in CAM particles during cycling, as shown by the *post mortem* TEM analysis in this paper. On the other hand, polycrystalline coatings will contain of grain boundaries, crystallographic defects, and spatial variations in topology and grain orientation. At the nanoscale, all of these factors will introduce local current focusing 'hot spots', which decreases the rate capability and stability of the cathode.

To clarify these points, the following text in the 'Introduction' section has been modified and highlighted:

Ideally, the coating would be chemically and structurally homogenous to avoid local current focusing at 'hot spots' such as grain/phase boundaries, crystallographic defects, and spatial variations in chemical composition, topology, and grain orientation of the coating. This last requirement is often overlooked in the design of artificial SEI/CEI layers; however, it has recently been shown that amorphous coatings have the potential to enable fast charging rates, which is attributed to a more uniform distribution of interfacial kinetics and transport compared to the 'natural' SEI layer that forms based on electrolyte decomposition^{28,29}. Hot spots can also arise if the coating is not perfectly conformal and continuous (pinhole-free), which requires precise synthesis methods such as atomic layer deposition (ALD)^{13,28,30}. Finally, the coating must be sufficiently mechanically compliant to withstand the cyclic strains that occur in CAM particles during cycling.

To further prove the benefits of using an amorphous-phase coating over the traditional polycrystalline LiNbO_3 coatings, we have performed new experiments comparing our ALD-coated cathode particles to those using the state-of-the-art solution processed LiNbO_3 coatings. We deposited polycrystalline LiNbO_3 coatings on the sc-NMC532 powders used in this study. The LiNbO_3 coating was applied using a solution processing method that includes a high-temperature annealing step at 425 °C, resulting in a polycrystalline coating^{5,33,41}.

The rate capabilities of composite cathodes having amorphous ALD Nb_2O_5 , and crystalline LiNbO_3 coatings at 4.3 V and 4.7 V are compared in Figure R2 below and the same has been added to the SI as Figure S11. The comparison clearly shows that although the crystalline LiNbO_3 -coated cathodes perform better than the uncoated cathode, they are still significantly inferior to the amorphous Nb_2O_5 -coated cathodes. For example, at 2C with a 4.3 V cutoff, the accessible capacities were 0.22, 1.16, and 1.64 $\text{mAh}\cdot\text{cm}^{-2}$, respectively for uncoated, LiNbO_3 , and Nb_2O_5 -coated cathodes. When the rate capability is performed at a higher cutoff voltage (4.7 V), the difference between LiNbO_3 and Nb_2O_5 electrodes gets further contrasted. With a 4.7 V cutoff, the LiNbO_3 -coated cathode showed a significant capacity loss, retaining only 0.45 $\text{mAh}\cdot\text{cm}^{-2}$ but the accessible capacity of the Nb_2O_5 -coated cathode remained fairly stable with a 1.49 $\text{mAh}\cdot\text{cm}^{-2}$. The performance of the uncoated cathode almost vanished.

To clarify, the following Figure has been added to the SI as Figure S11:

Figure R2: Rate capability trends of uncoated, ALD Nb₂O₅ coated, and solution-processed LiNbO₃ coated sc-NMC composite cathodes at (A) 4.3 V and (B) 4.7 V cutoff voltages.

To clarify, the following text has been added to the main text:

To provide a comparison to state-of-the-art coatings, solution-processed LiNbO₃ coatings were also applied to sc-NMC cathodes^{5,33,41}. The rate capability of ALD Nb₂O₅-coated cathodes cycled to 4.3 V and 4.7 V cutoff voltages is consistently higher than those of LiNbO₃-coated cathodes. It is important to highlight that the solution processing method involves a high-temperature annealing step, resulting in a polycrystalline coating with microstructural heterogeneity. The superior performance of the Nb₂O₅ coating underscores the benefits of an amorphous coating.

To clarify, the following text has been added to the ‘Methods’ section:

Solution-processed LiNbO₃ coating on sc-NMC532 powders:

A 3 wt% LiNbO₃ coating was deposited on sc-NMC532 powders using the solution processing method, following the published literature^{5,33,41}. A solution of 5% (w/v) lithium niobium ethoxide in ethanol (Thermo Scientific Chemicals) was added to 0.5 g of sc-NMC532 powder, stirred for 1 h, and dried at 70 °C for 12 h under Ar flow. Finally, this dried powder was annealed in an oxygen atmosphere at 425 °C for 1.5 h, resulting in a LiNbO₃ coating on sc-NMC532 particles.

Comment 3: What irreversible structural and chemical changes occur in the uncoated cathode materials mentioned in the article under high voltage cycling?

Response: Thank you for the comments. We have shown and discussed multiple irreversible structural and chemical changes and their impact on the performance of the uncoated cathode during high-voltage cycling. For example, Figure 2G-I and Figure S13 highlight the evolution of higher interfacial impedance contributed by more CEI formation and irreversible structural changes. In Figure 5 and Figure S18, HRTEM images and XRD plots after cycling at different cutoff voltages reveal irreversible structural changes, highlighting the formation of a kinetically less active rock-salt phase. Furthermore, in Figure 6, lattice/planner gliding, and sub-surface and intra-particle cracking in uncoated NCM particles have been demonstrated.

For the reviewer's reference, these figures are provided below:

Figure 3: (G) Nyquist impedance plots of uncoated and Nb₂O₅-coated cathodes after the rate capability tests. (H) Zoomed-in view showing Nyquist impedance plots of Nb₂O₅-coated cathodes. (I) Comparison of interfacial impedance for uncoated and Nb₂O₅-coated cathodes after the rate capability tests.

Figure S13: (A) Intermittent EIS measurement at specific voltage points during the charging cycle (at C/10 rate) of uncoated and Nb₂O₅-coated cathodes against a Li-metal anode. Zoomed-in view of Nyquist plots

obtained at different voltage points during the charging cycle are presented for (B) uncoated and (D) Nb₂O₅-coated cathodes, with insets providing a zoomed-out view of the Nyquist plots. (C) Comparison of total interfacial impedance at different voltage points during charging.

Figure 5: (A) Schematic showing structural changes occurring in uncoated sc-NMC532 particles when cycled to different voltage limits. Upon repeated cycling to high voltage limits (>4.3 V), uncoated particles undergo irreversible structural changes (spinel and rock-salt phase formations), and (sub)surface and intraparticle cracking. HAADF-STEM images after 500 cycles at 1C rate for (B) uncoated cycled at 4.3 V and (C) uncoated cycled at 4.7 V are presented. FFT patterns and zoomed-in views of marked regions are also presented as insets. (D-F) *Ex-situ* XRD scans (zoomed-in) of uncoated samples before cycling obtained from samples after rate capability test at 4.3 V and 4.7 V limits.

Figure 6: (A) Schematic showing the onset of lattice gliding, subsequent microcrack formation, and its growth in (sub)surface and intra-particle cracking. (B) HAADF-STEM image of uncoated sc-NMC particle after 500 cycles at 1C rate and 4.7 V shows serrated surface resulting from lattice gliding and (sub)surface cracking. (C) Zoomed-in images of the cracks highlighted by white box in (B) and FFTs from different marked regions. FIB-SEM cross-section images of uncoated sc-NMC electrode after 500 cycles at 1C rate and 4.7 V showing (D) (sub)surface and (E) intra-particle cracks.

Comment 4: In the manuscript, the authors mention that it is the first time that Nb₂O₅ is used as a coating material for SSB cathodes with improved efficiency and cycling stability. On what technique were these results realized compared to other studies?

Response: We thank the Reviewer for the comment. To the best of our knowledge, this is the first demonstration of fabrication of a pin-hole free conformal niobium oxide coating by ALD method for SSB cathodes. The coatings have a Nb:O atomic ratio of 2:5 (i.e., Nb₂O₅) and amorphous phase (Figure 1B,C and Figure S4). We confirmed its composition by multiple techniques (XPS and TEM-EDS) (Table S1 and Figure S2). The ALD process allowed precisely tuning the composition while low-temperature processing (175 °C) enabled amorphous nature, preventing the formation of crystalline phase and any Nb-doping to NMC (Figure S6).

To emphasize the novel aspects of our methods compared to previous studies, we revised the title of the paper as follows:

Eliminating chemo-mechanical degradation of solid-state battery cathodes during >4.5V cycling using amorphous ALD coatings

On the other hand, the majority of prior reports to date on niobium-based coatings have used different methods including wet-chemical, ALD followed by heat treatment, or solid-state reactions. Importantly, these techniques have typically included a high-temperature annealing step either during or after the coating synthesis on the cathode particles, which leads to the formation of crystalline Li_xNbO_y. We have provided a comprehensive survey of different coating materials for high-voltage cathodes, their synthesis methods, and electrochemical performance in Table S4 in the SI.

For your reference, Table S4 is revisited below.

Table S4: Comparison of electrochemical performance of layered cathode materials having different coatings in solid electrolyte and liquid electrolyte systems.

Coating material	Coating Thickness	Coating method	Cathode material	Electrolyte	Anode	Upper Voltage limit (V vs Li/Li ⁺)	1 st cycle CE (Voltage limit, current density)	Capacity retention as % (voltage limit, C-rate, after cycles)	Ref
Al ₂ O ₃	0.4-1.4 nm	ALD	LiNi _{0.5} Mn _{1.5} O ₄	Li ₆ PS ₅ Cl	Li-In	4.4V (5.0V)	86.5% (5.0V, 0.2C)	70.1% (5.0V, 0.2C, 100 cycles)	²
ZrO ₂	4-5 nm	ALD	LiNi _{0.85} Co _{0.1} Mn _{0.5} O ₂	Li ₆ PS ₅ Cl	LTO	2.75V (4.3V)	~91% (4.3V, 0.1C)	78% (4.3V, 0.5C, 200 cycles)	³
HfO ₂	2-3nm	ALD	LiNi _{0.85} Co _{0.1} Mn _{0.5} O ₂	Li ₆ PS ₅ Cl	Li	4.3V	~88% (4.3V, 0.1C)	82% (4.3V, 0.5C, 60 cycles)	⁴
LiNbO ₃	2-5 nm	Solution	LiNi _{0.82} Co _{0.12} Mn _{0.6} O ₂	Li ₆ PS ₅ Cl	Li-In	3.7V (4.32) 3.9V (4.52V)	71.8% (4.32V, 8.5 mA/g)	82.1% (4.52V, 34 mA/g, 30 cycles)	⁵

LiNbO ₃	10-20nm	Solution	LiNi _{0.5} Mn _{1.5} O ₄	Li ₆ PS ₅ Cl/ Li ₃ YCl ₆	Li-In	4.25V (4.85V)	91.2% (4.85V, 7.5 mA/g)	~50% (4.85V, 20mA/g, 50 cycles)	6
LiNbO _x	4 nm	ALD	LiNi _{0.8} Co _{0.1} Mn _{0.1} O ₂	Li ₁₀ GeP ₂ S ₁₂	LTO	2.8V (4.35V)	80.6% (4.35V, 0.1C)	76.3% (4.35V, 0.3C, 400 cycles)	7
Li ₃ BO ₃	1-11 nm	Solution	LiCoO ₂	Li ₆ PS ₅ Cl	Li-In	3.68V (4.3V) 3.88V (4.5V)	91% (4.3V)	88.7% (4.5V, 0.2C, 25cycles)	8
Li ₃ BO ₃ - Li ₂ CO ₃	21-30 nm	Solution	LiCoO ₂	Li ₆ PS ₅ Cl	Li-In	3.68V (4.3V) 3.88V (4.5V)	93% (4.3V)	93.8% (4.5V, 0.2C, 25cycles)	8
LiTaO _x	2-6 nm	Solution	LiNi _{0.82} Co _{0.12} Mn _{0.6} O ₂	Li ₆ PS ₅ Cl	Li-In	3.7V (4.32) 3.9V (4.52V)	76.1% (4.32V, 8.5 mA/g)	83% (4.52V, 30 cycles)	5
Li ₃ PO ₄	1-10 nm	ALD	LiNi _{0.8} Co _{0.1} Mn _{0.1} O ₂	Li ₁₀ GeP ₂ S ₁₂	In	3.88V (4.5V)	75.1% (4.5V, 0.1C)	78% (4.4V, 0.2C, 100 cycles)	9
Li ₂ ZrO ₃	<10 nm	Solution	LiNi _{0.82} Co _{0.12} Mn _{0.6} O ₂	Li ₆ PS ₅ Cl	LTO	2.85V (4.4V)	86% (4.4V, 0.2C)	~70% (4.4V, 0.1C, 60 cycles)	10
LiWO ₃	2-4 nm	Solution	LiNi _{0.6} Co _{0.2} Mn _{0.2} O ₂	75Li ₂ S- 22P ₂ S ₅ - 3Li ₂ SO ₄	Li-In	3.88V (4.5V)	64.4% (4.5V, 0.05C)	83% (4.5V, 0.1C, 30 cycles)	11
Li _x Al _y Zn _z O ₅	~4 nm	ALD	LiNiO ₂	Li ₆ PS ₅ Cl	Li-In	4.3V	85.4% (4.3V, 0.2C)	83.1% (4.3V, 0.2C, 200 cycles)	12
LiAl(PO ₃) ₄	4nm	ALD	LiNi _{0.88} Co _{0.09} Mn _{0.03} O ₂	Li ₆ PS ₅ Cl	Li-In	4.3V	84.1%, 4.3V, C/5	98.3%, 440 cycles 20.1 mg/cm ²	13
CoO/Li ₂ CO ₃	4nm	Heat treatmen t	LiCoO ₂	Li ₆ PS ₅ Cl	Li-In	4.6V	83%, 4.3V, C/2	83%, 150 cycles, C/2	14
Gd ₂ O ₃	7 nm	Solution	LiNi _{0.6} Co _{0.05} Mn _{0.35} O ₂	Liquid	Li	4.5V	~83% (4.5V, 0.1C)	88.1% (4.5V, 1C, 400 cycles)	15
Sm ₂ O ₃	13 nm	Solution	LiNi _{0.6} Co _{0.05} Mn _{0.35} O ₂	Liquid	Li	4.5	~82% (4.5V, 0.1C)	97.0 % (4.5V, 1C, 300 cycles)	16
Al ₂ O ₃	1-4 nm	ALD	LiNi _{0.6} Co _{0.2} Mn _{0.2} O ₂	Liquid	Li	4.7V	~85%(4.7 V, 0.5C)	89.5% (4.7V, 0.5C, 45 cycles)	17
ZrO ₂	Not available	Ball mill	Li _{1.2} Ni _{0.13} Co _{0.13} Mn _{0.54} O ₂	Liquid	Li	4.8V	82.5 (4.8V, 0.1C)	89.0% (4.8V, 1C, 100 cycles)	18
Li ₃ PO ₄	20 nm	Solution	LiCoO ₂	Liquid	Li	4.5V	87.3% (4.5V, 0.1C)	90% (4.5V, 0.5C, 100 cycles)	19
AlZnO	3nm	solution	LiCoO ₂	Liquid	Li	4.6V	~82% (4.6V, 37 mA/g)	65.7% (4.6V, 185 mA/g, 500 cycles)	20

Li _{0.5} Mn _{0.5} O		solution	Li _{1.2} Mn _{0.6} Ni _{0.2} O ₂	Liquid	Li	4.8V	80.3% (4.8V, C/10)	80.7% after 200 cycles at 1 C	²¹
--	----------	--	--------	----	------	--------------------------	-------------------------------------	---------------

Comment 5: “Composite cathodes consisting of uncoated and Nb₂O₅-coated sc-NMC particles were prepared by mixing Li₆PS₅Cl (as the SE), graphitized carbon nanofibers (as the conducting additive), and PTFE (as the binder) in a ratio of 70:30:5:5.” Is such a mixing ratio supported by the literature? Also, the proportions of the three materials are incorrectly indicated.

Response: We thank the Reviewer for raising this point. The composition of an SSB cathode greatly affects its microstructure, and electronic and ionic percolation pathways, critically influencing the energy and power densities. To provide a balance between energy and power density, composite SSB cathodes are often composited cathode active material and solid electrolyte in a weight ratio of approximately 70:30, in addition to other phases (conducting additive and binder) that has been widely used in the literature [*Commun. Mater.* **2** (2021) 112; *Chem. Mater.* **33** (2021) 2624–2634; *Batteries & Supercaps* **5** (2022) e202100397].

We conducted a systematic study on the effect of cathode composition on the rate performance of the uncoated cathode (Figure R3) and chose the 70:30:5:5 composition. This convention of expressing the relative loadings is commonly used in the literature; however, we understand the Reviewer’s point that an alternative expression may be to express then as a percentage, rather than a ratio. Therefore, we have now clarified the corresponding weight ratio of this mixture:

To clarify, the following text has been modified in the main text:

Composite cathodes consisting of uncoated and Nb₂O₅-coated sc-NMC particles were prepared by mixing Li₆PS₅Cl (as the SE), graphitized carbon nanofibers (as the conducting additive), and PTFE (as the binder) in a weight ratio of 70:30:5:5 (or weight % ratio of 63.6:27.3:4.6:4.6).

To understand the role of mixing ration on performance, we performed an optimization study of cathodes with varying ratios, as shown in Fig. R3 below. The results showed that the cathodes with higher solid electrolyte content (e.g., 30:70:5:5 or 50:50:5:5) exhibited greater accessible capacity and high-rate capability. In contrast, the cathodes with higher active material content (e.g., 80:20:5:5 and 90:10:5:5) experienced more significant rate limitations.

Figure R3: Effect of cathode composition on rate capability

To clarify, the following text has been modified in the 'Methods' section:

For composite cathode preparation, NMC powders (uncoated or coated; Li-capacity 165 mAh.g⁻¹ at 4.3V), Li₆PS₅Cl SE phase (≤1μm size; MSE Supplies), PTFE binder (Sigma Aldrich) and graphitized carbon nanofibers conductive additive (Sigma Aldrich) were mixed in a weight ratio of 70:30:5:5 (or weight % ratio of 63.6:27.3:4.6:4.6).

Comment 6: What are the results of the rate capability tests of the composite positive electrode at different voltage limits mentioned in the article, and how does the Nb₂O₅ coating affect this performance?

Response: The rate capability results from uncoated and Nb₂O₅-coated cathodes, cycled to different cutoff voltages (4.3 V, 4.5 V, 4.7 V), including accessible capacities, impedance (before and after the rate capability tests), and polarization (derived from the dQ/dV plots), were presented in Figure 3, Figure S7 and Figure S9 (now Figure S9 and Figure S12 in the revised SI). These results have been discussed in 'Effect on Rate Capability' and 'Effect on Cell Polarization and Impedance Evolution' sections.

For the Reviewer's reference, we are revisiting these figures below:

Figure 3: Rate capability trends of (A) uncoated and (B) Nb₂O₅ coated composite cathodes (vs. LTO anode). Voltage profiles of (C) uncoated and (D) Nb₂O₅ coated cathodes at different c-rates (C/10, C/5, C/2, 1C, 2C) at a cutoff voltage of 4.7V. Comparison of polarization estimated from dQ/dV analysis of voltage profiles at different c-rates for (E) uncoated and (F) Nb₂O₅ coated electrodes. (G) Nyquist impedance plots of uncoated and Nb₂O₅-coated cathodes after the rate capability tests. (H) Zoomed-in view showing Nyquist impedance plots of Nb₂O₅-coated cathodes. (I) Comparison of interfacial impedance for uncoated and Nb₂O₅-coated cathodes after the rate capability tests.

Figure S9: Voltage profiles at different c-rates (C/10, C/5, C/2, 1C, 2C) for (A-C) uncoated and (D-F) Nb₂O₅ coated sc-NMC composite cathodes cycled at different cutoff voltages.

Figure S12: dQ/dV vs V plots obtained from the voltage profiles during rate capability tests of (A-C) uncoated and (D-F) Nb₂O₅ coated sc-NMC composite cathodes cycled at different cutoff voltages.

Comment 7: There are a large number of abbreviations in this paper that make reading and understanding of texts difficult. The parameters should be introduced in appendix section, their descriptions are omitted from the paper.

Response: We thank the Reviewer for this suggestion. For more clarity and better readability, we now summarized the abbreviations in Table S5 in the SI.

To clarify, the following table has been added to the SI as Table S5:

Table S5: List of abbreviations used in the study

SSB: Solid-state battery	SEI: Solid electrolyte interface
LIB: Li-ion battery	ASR: Area-specific resistance
EV: Electric vehicle	XPS: X-ray photoelectron spectroscopy
SE: Solid electrolyte	FIB: Focused ion beam
LE: Liquid electrolyte	TEM: Transmission electron microscopy
CAM: Cathode active material	STEM: Scanning transmission electron microscopy
NMC: Nickel manganese cobalt oxide	EDS: Electron dispersive X-ray spectroscopy
SC: single crystal	HAADF: High-angle annular dark-field
LPSC: $\text{Li}_6\text{PS}_5\text{Cl}$	XRD: X-ray diffraction
ALD: Atomic layer deposition	EIS: Electrochemical impedance spectroscopy
TM: Transition metal	GITT: Galvanostatic intermittent titration technique
PTFE: Polytetrafluoroethylene	FFT: Fast Fourier transform
LTO: Lithium titanate	CC: Constant current
CEI: cathode electrolyte interface	CV: Constant voltage
CE: Coulombic efficiency	CCCV: Constant current constant voltage
ICE: Initial Coulombic efficiency	

Comment 8: The analysis of the initial Coulomb efficiency was not specific enough. The results in Figure 2 are not sufficient to verify the high efficiency and stability of the Nb_2O_5 coating.

Response: We thank the Reviewer for the comment, and would like to discuss the initial Coulombic efficiency (ICE) further. Figure 2 presents the average ICE from three sets of samples at each condition with error bars, demonstrating the repeatability of our results and the superior performance of Nb_2O_5 -coated samples starting from the formation cycling. The high ICE significantly results in improved rate capability (at high current densities) and long-term cycling stability.

To provide further analysis of ICE, we have now added new analysis of the detailed voltage traces and corresponding dQ/dV plots during the first formation cycles of the cathode samples cycled to different cutoff voltages (4.3 V, 4.5 V, 4.7 V) in Figure R4 below. For the uncoated samples, the voltage traces appear more sloped and reach the cutoff voltage early during the discharge cycle. On the other hand, the voltage traces of Nb_2O_5 -coated samples appear relatively less sloped and take more time to reach the cutoff voltage during the discharge cycle. Additionally, the uncoated samples exhibit higher higher cell polarization during the discharge cycles compared to the coated samples. Furthermore, the dQ/dV plots show inferior reversibility for the uncoated samples and greater accessibility and reversibility for the coated samples.

Figure 2: Initial Coulombic efficiency of uncoated and Nb₂O₅ coated cathodes cycled to different cutoff voltages during the first formation cycle.

To clarify, the following Figure has been added in the SI as Figure S8:

Figure R4: First formation cycle voltage profiles and corresponding dQ/dV vs V plots at different cutoff voltages for (A,C) uncoated and (B,D) Nb₂O₅ coated sc-NMC composite cathodes, respectively.

To clarify, the following text has been added to the main text:

The voltage traces during the first formation cycle of the uncoated samples exhibit higher cell polarization and appear more sloped, reaching the cutoff voltage earlier during the discharge cycle (Figure S8).

Comment 9: There are a large number of yellow-marked paragraphs in the main text content, which is not a professional representation.

Response: We apologize for the confusion caused due to the yellow-marked paragraphs in the manuscript provided. These yellow highlights were responses to previous reviewer comments from an earlier submission.

Comment 10. The fonts of several images in the manuscript should all be appropriately enlarged to enhance readability.

Response: Thank you for your feedback. We have appropriately enlarged the fonts in the figures in the revised manuscript to enhance readability and overall clarity.

Thank you for your consideration of this article for publication.

Sincerely,

Neil Dasgupta
Associate Professor, Miller Faculty Scholar
Department of Mechanical Engineering
Department of Materials Science & Engineering
University of Michigan, Ann Arbor

We thank the Reviewer for their positive feedback. We have carefully addressed all comments and made necessary changes to the manuscript that are highlighted in the revised manuscript, SI, and response letter below. Thank you again for your consideration of this manuscript for publication in *Nature Communications*.

Reviewer #3:

The authors have revised the content of the manuscript in detail compared to the previous one, and the necessary analysis has enhanced the quality and rigor of the article. Therefore, this study on the stability and performance enhancement of solid-state batteries (SSBs) deserves to be published, provided that minor revisions are made to the following details. The specific comments are as follows.

Response: We thank the reviewer for the positive feedback, recognizing our efforts and enhanced quality of the revised manuscript and recommending it for publication in *Nature Communications*. Below, we provide point-by-point responses to the specific questions.

Comment 1: The first occurrence of a term in the abstract should be explained in full, and the full name should be indicated in the SE to increase readability for the reader.

Response: We have now defined SE at its first occurrence in the abstract.

Comment 2: In the abstract, “Compared to uncoated samples at high voltages (≥ 4.5 V), the composite cathode with Nb₂O₅-coated CAM particles demonstrates a high initial Coulombic efficiency (91% vs. 82%),” the comparative representation of Coulombic efficiency in parentheses seems inappropriate, and a more standardized expression is suggested.

Response: We thank the Reviewer for the suggestion. We have revised the statement to ensure scientific clarity.

To clarify the following text has been modified in the abstract:

At high-voltages (≥ 4.5 V), the composite cathode with Nb₂O₅-coated CAM particles demonstrates a higher initial Coulombic efficiency of 91% compared to 82% for the uncoated samples, along

with improved rate capability (10x higher accessible capacity at 2C rate) and remarkable capacity retention during extended high-voltage cycling (99.4% after 500 cycles at 4.7 V).

Comment 3: Units appearing in the manuscript should be examined in detail. For example, should $2 \text{ mS}\cdot\text{cm}^{-1}$ be revised to $2 \text{ mS}\cdot\text{cm}^{-1}$, and should $>275 \text{ mAh}\cdot\text{g}^{-1}$ be revised to $>275 \text{ mAh}\cdot\text{g}^{-1}$?

Response: We have revised the format of the units throughout the main text and SI.

Comment 4: “Recently, it has been shown that mechanical degradation (e.g., intergranular cracking) can be reduced by using single crystal (sc) NMC particles”. Why is the abbreviation single crystal used here with a lowercase sc?

Response: We have replaced this abbreviation with uppercase SC throughout the main text and SI.

Comment 5: “This last requirement is often overlooked in the design of artificial SEI/CEI layers; however, it has recently been shown that amorphous coatings have the potential to enable fast charging rates”. This sentence suggests a change.

Response: We thank the Reviewer for the feedback. This sentence has been modified as follows:

The importance of chemical and structural homogeneity of the coating is often overlooked in the design of artificial SEI/CEI layers. However, recent studies have shown that amorphous coatings can enable fast-charging capabilities, which is attributed to a more uniform distribution of interfacial kinetics and transport compared to the composite ‘natural’ SEI layer that forms based on electrolyte decomposition^{28,29}

Comment 6: “The composite SSB cathodes containing Nb₂O₅-coated sc-NMC particles, along with LPSC solid electrolyte, binder, and conducting additive, show significantly improved electrochemical performance under high voltage cycling ($\geq 4.5 \text{ V}$) including initial Coulombic efficiency (CE: 91% vs. 83%)” where sc and CE; 91% vs. 83% are suggested to be modified.

Response: This sentence is now modified in the introduction, as provided below:

The composite SSB cathodes containing Nb₂O₅-coated SC-NMC particles show significantly improved electrochemical performance under high-voltage cycling ($\geq 4.5 \text{ V}$) including a higher initial Coulombic efficiency of 91% compared to 82% for the uncoated samples, improved rate capability (10x higher accessible capacity at 2C rate), and long-term cycling stability (99.4% after 500 cycles) compared to uncoated SSB cathodes.

Comment 7: The font size in Figure 4 is not consistent. For example, LTO/SE/NMC.

Response: We have corrected and revised the figure in the main text.

Sincerely,

Neil Dasgupta

Associate Professor, Miller Faculty Scholar

Department of Mechanical Engineering

Department of Materials Science & Engineering

University of Michigan, Ann Arbor